# Thermodynamically Consistent Latent Dynamics Identification for Parametric Systems

**Xiaolong He**                                                     *xiaolong.he@synopsys.com*
*Synopsys Inc.*

**Yeonjong Shin**                                                   *yeonjong_shin@ncsu.edu*
*Department of Mathematics*
*North Carolina State University*

**Anthony Gruber**                                                  *adgrube@sandia.gov*
*Center for Computing Research*
*Sandia National Laboratories*

**Sohyeon Jung**                                                    *sjung55@asu.edu*
*School of Mathematics and Statistical Sciences*
*Arizona State University*

**Kookjin Lee**                                                     *kookjin.lee@asu.edu*
*School of Computing and Augmented Intelligence*
*Arizona State University*

**Youngsoo Choi**                                                   *choi15@llnl.gov*
*Center for Applied Scientific Computing*
*Lawrence Livermore National Laboratory*

**Reviewed on OpenReview:** *https://openreview.net/forum?id=Qy3oLpRzpf*

## Abstract

We propose an efficient thermodynamics-informed latent space dynamics identification (tLaSDI) framework for the reduced-order modeling of parametric nonlinear dynamical systems. This framework integrates autoencoders for dimensionality reduction with the newly developed parametric GENERIC formalism-informed neural networks (pGFINNs), which enable efficient learning of parametric latent dynamics while preserving key thermodynamic principles, such as free energy conservation and entropy generation, across the parameter space. To further enhance model performance, a physics-informed active learning strategy is incorporated, leveraging a greedy, residual-based error indicator to adaptively sample informative training data, outperforming uniform sampling at equivalent computational cost. Numerical experiments on the Burgers' equation and the 1D/1V Vlasov-Poisson equation demonstrate that the proposed method achieves up to $2,495\times$ speed-up over the full-order numerical baseline with 1-3% relative errors, as well as significant reductions in training (50-90%) and inference (57-61%) cost. Moreover, the learned latent space dynamics reveal the underlying thermodynamic behavior of the system, offering valuable insights into the physical-space dynamics. Code is available at the repository: `https://github.com/xiaolong7/pGFINN-tLaSDI`.

## 1 Introduction

Understanding and simulating nonlinear dynamical systems is fundamental to advancing numerous fields in science and engineering, from physics (Oberkampf & Trucano, 2002; Thijssen, 2007) and electromagnetics

(Bondeson et al., 2012) to material sciences (Lee, 2016) and digital twin technologies (Jones et al., 2020; Liu et al., 2021). However, high-fidelity numerical models for such systems are often computationally prohibitive, especially in real-time or many-query settings such as optimization (Sigmund & Maute, 2013; White et al., 2020), uncertainty quantification (Galbally et al., 2010; Abdar et al., 2021), or digital twin deployment (Jones et al., 2020; Liu et al., 2021). Reduced-order models (ROMs) offer a promising solution to this challenge through their identification of low-dimensional representations that capture the essential dynamics of the system.

Traditional ROMs based on linear projection techniques, such as proper orthogonal decomposition (POD) (Berkooz et al., 1993) and reduced basis methods (Patera & Rozza, 2007), have demonstrated great promise, but typically require direct access to high-fidelity numerical solvers (McBane & Choi, 2021). Additionally, they often struggle with advection-dominated or highly nonlinear systems, characterized by the slow decay of their Kolmogorov $n$-width, which quantifies the best-case approximability of the system state using a linear subspace of dimension $n$ (Fries et al., 2022). To address these limitations, nonlinear manifold learning via autoencoders (Hinton & Salakhutdinov, 2006) has been explored, offering enhanced expressiveness for capturing complex solution manifolds (Fulton et al., 2019; Lee & Carlberg, 2020; Kim et al., 2022; Maulik et al., 2021; Lee & Parish, 2021) at the cost of greater computational complexity. Recent advancements further integrate equation learning methods (Schmidt & Lipson, 2009; Brunton et al., 2016; Peherstorfer & Willcox, 2016; Messenger & Bortz, 2021b;a) with latent space embeddings to discover interpretable latent dynamics which can be simpler and easier to learn than their physical-space counterparts (Champion et al., 2019; Qian et al., 2020; Benner et al., 2020; Fries et al., 2022; He et al., 2023; Bonneville et al., 2024a; Tran et al., 2024; He et al., 2025; Anderson et al., 2025).

Despite these advancements, many existing data-driven ROMs fail to enforce fundamental physical principles, such as energy conservation or the second law of thermodynamics, which are responsible for dynamical stability and long-term system behavior. As a result, their latent dynamics may fit a given set of training data well while exhibiting unphysical behavior under extrapolation or unseen conditions. To bridge this critical gap, thermodynamic priors have been introduced into data-driven ROMs. Yu et al. Yu et al. (2021) introduced an OnsagerNet to learn stable and interpretable dynamics of dissipative systems by enforcing a generalized Onsager principle (Onsager, 1931). Geng et al. Geng et al. (2025) extended Hamiltonian operator inference (Sharma et al., 2022; Gruber & Tezaur, 2023) to port-Hamiltonian systems, inferring reduced operators from state/output snapshots and a chosen Hamiltonian ansatz to yield ROMs that preserve energy storage, dissipation, and input–output passivity. Park et al. Park et al. (2024) introduced a thermodynamics-informed latent space dynamics identification (tLaSDI) framework, which integrates nonlinear manifold learning with GENERIC (General Equation for Non-Equilibrium Reversible-Irreversible Coupling (Grmela & Öttinger, 1997; Öttinger, 2005)) formalism-informed neural networks (GFINNs) (Zhang et al., 2022) to learn latent dynamics that are consistent with thermodynamical laws. In tLaSDI, the effective parameterization of latent dynamics is achieved through a hyper-autoencoder architecture, where a hypernetwork (Ha et al., 2017) generates autoencoder coefficients based on input parameters. While this design enables flexible parameterization, it also introduces significant model complexity and training overhead, posing challenges for scalability in large-scale or high-dimensional parametric settings.

To address this issue, this work proposes a novel and efficient tLaSDI framework for parametric nonlinear dynamical systems. Our approach integrates standard autoencoders for nonlinear dimensionality reduction with newly developed parametric GFINNs (pGFINNs). Besides incorporating parametric dependence in a simple and interpretable way, these networks are specifically designed to enforce the GENERIC structure, ensuring that the latent dynamics conserve free energy and generate entropy consistently across the parameter space, thus satisfying both the first and second laws of thermodynamics. To further enhance model performance, we incorporate a physics-informed active learning strategy using a greedy, residual-based error indicator that adaptively selects the most informative parameter samples during training. This targeted sampling strategy enables effective generalization across parameter space, outperforming uniform sampling at the same computational cost.

The proposed tLaSDI framework is validated through numerical experiments on the Burgers' equation and the 1D/1V Vlasov–Poisson equation, which represent standard nonlinear benchmarks for model reduction and dynamics discovery. Compared to the hyper-autoencoder-based parameterization, our method reduces

training cost by 50%-90% and inference time by 57%-61%, enabling superior accuracy and interpretability at a greatly reduced computational cost. Furthermore, for the Vlasov–Poisson system, we achieve up to $2,495\times$ speed-up with less than 3% relative errors in a predictive setting, when compared to a traditional numerical simulation that produces a similar error with respect to the high-fidelity training data. Beyond these computational gains, our framework provides physically interpretable latent variables that reflect the system's thermodynamic behavior and offer insights into its evolution in both the latent and physical spaces. For example, the learned entropy in the Vlasov–Poisson system is seen to correspond to the growth rate in the two-stream instability, providing a physically relevant measure useful for further analysis.

In summary, this work represents a significant step toward the data-efficient, interpretable, and physically-consistent reduced-order modeling of complex parametric systems, and positions the proposed tLaSDI as a robust foundation for next-generation ROMs that are both scientifically principled and computationally scalable. Our principal contributions are:

- the development of pGFINNs to identify thermodynamically consistent latent dynamics enforcing free energy conservation and entropy generation across varying system parameters;
- the proposal of an efficient tLaSDI framework for reduced-order modeling through the integration of standard autoencoders with the developed pGFINNs;
- the integration of an active learning strategy into the proposed tLaSDI framework for adaptive parameter sampling on-the-fly, leading to improved generalization capabilities across parameter space at equivalent cost to uniform sampling.
- the production of interpretable and physically-relevant latent dynamics, achieving up to $2,495\times$ speed-up with 1-3% errors on parametric benchmarks, as well as significant reductions in training (50-90%) and inference (57-61%) cost.

The remainder of this work details these contributions and their consequences. Additional implementation details necessary for reproducing the results in the body can be found in Appendix.

## 2 Methodology

We consider a parametric dynamical system characterized by a system of ordinary differential equations (ODEs)

$$\dot{\mathbf{u}}(t;\boldsymbol{\mu}) = \mathbf{f}(\mathbf{u}, t; \boldsymbol{\mu}), \quad t \in [0, T], \quad \mathbf{u}(0; \boldsymbol{\mu}) = \mathbf{u}_0(\boldsymbol{\mu}) \in \mathbb{R}^{N_u}, \tag{1}$$

where $\mathbf{u}(t;\boldsymbol{\mu})$ represents the parametric, time-dependent solution to the dynamical system, $T \in \mathbb{R}_+$ is the final time, $\mathbf{f}$ is an appropriate field function that guarantees the existence of the solution, and $\mathbf{u}_0$ is the initial state of $\mathbf{u}$, parametrized by $\boldsymbol{\mu} \in \mathcal{D} \subseteq \mathbb{R}^{N_\mu}$. Eq. (1) can be considered as a semi-discretized system of partial differential equations (PDE), in which the numerical solution to Eq. (1) can be obtained using explicit or implicit time integration schemes. For example, with the implicit backward Euler time integrator, we compute an approximate solution by solving the nonlinear system of equations $\mathbf{u}_n = \mathbf{u}_{n-1} + \Delta t\, \mathbf{f}_n$, where $\mathbf{u}_n \approx \mathbf{u}(t_n; \boldsymbol{\mu})$ is the approximate state and $\mathbf{f}_n := \mathbf{f}(\mathbf{u}_n, t_n; \boldsymbol{\mu})$ is the discrete velocity field. The residual function is then expressed as

$$\mathbf{r}(\mathbf{u}_n; \mathbf{u}_{n-1}, \boldsymbol{\mu}) = \mathbf{u}_n - \mathbf{u}_{n-1} - \Delta t\, \mathbf{f}_n. \tag{2}$$

Computing the time integration corresponding to Eq. (2) is often computationally expensive, especially when the ambient dimension of the solution $N_u$ is large. In this work, we aim to develop an accurate and efficient ROM framework based on thermodynamically consistent latent space dynamics identification, which preserves the thermodynamic character of the physics described by Eq. (1) in a machine-learned latent space where the governing dynamics have been distilled. The remainder of this section describes the proposed pGFINN architecture, its integration with tLaSDI, and the physics-informed active learning strategy which enables improved generalization performance.

### 2.1 Parametric GFINN (pGFINN)

The GENERIC formalism (Grmela & Öttinger, 1997; Öttinger, 2005), also known as the metriplectic formalism (Morrison, 1986; 2009), is a mathematical framework describing beyond-equilibrium thermodynamic

systems, including both conservative and dissipative systems. Its equations of motion are compactly expressed as

$$\dot{\mathbf{z}} = \boldsymbol{L}(\mathbf{z})\frac{\partial E}{\partial \mathbf{z}}(\mathbf{z}) + \boldsymbol{M}(\mathbf{z})\frac{\partial S}{\partial \mathbf{z}}(\mathbf{z})$$

$$\text{subject to} \quad \boldsymbol{L}(\mathbf{z}) = -\boldsymbol{L}(\mathbf{z})^T, \quad \boldsymbol{M}(\mathbf{z}) = \boldsymbol{M}(\mathbf{z})^T \succeq \mathbf{0}, \tag{3}$$

$$\boldsymbol{L}(\mathbf{z})\frac{\partial S}{\partial \mathbf{z}}(\mathbf{z}) = \boldsymbol{M}(\mathbf{z})\frac{\partial E}{\partial \mathbf{z}}(\mathbf{z}) = \mathbf{0}$$

where $\mathbf{z} \in \mathbb{R}^d$ denotes the state variable, $E$ resp. $S$ are scalar functions representing the total energy resp. entropy of the system, and $\boldsymbol{L}$ resp. $\boldsymbol{M}$ are matrix-valued functions representing the skew-symmetric Poisson resp. symmetric positive semi-definite friction matrices. The final set of conditions prescribing the kernels of $\boldsymbol{L}$ and $\boldsymbol{M}$ can be shown to instantaneously guarantee the first two laws of thermodynamics, i.e., energy conservation $\dot{E}(\mathbf{z}) = 0$ and non-decreasing entropy $\dot{S}(\mathbf{z}) \geq 0$ (Gruber et al., 2023a). Due to these general and favorable properties, the model reduction and dynamics identification of GENERIC/metriplectic systems is an active area of development, including strategies such as (Gruber et al., 2023a; Lee et al., 2021; Zhang et al., 2022; Gruber et al., 2023b; 2025). Recently, Zhang et al. proposed GFINNs (Zhang et al., 2022), designed to exactly satisfy the degeneracy conditions in Eq. (3) through structure-informed architectures for the governing data $\boldsymbol{L}, \boldsymbol{M}, E, S$. However, GFINNs do not natively accommodate differential equations with parametric variability in their governing operators. In this work, we introduce parametrization into GFINNs to enhance their ability to handle such dynamical systems, ensuring that the learned solutions remain accurate in these situations.

To minimize the requirements of this approach, we consider the most general setting where no prior knowledge on the system Eq. (1) is assumed. Therefore, all data defining the GENERIC surrogate are approximated with carefully constructed neural networks (NNs), i.e. $E_{\text{NN}}$, $S_{\text{NN}}$, $\boldsymbol{L}_{\text{NN}}$, and $\boldsymbol{M}_{\text{NN}}$. The architecture of pGFINN is illustrated in Fig. 1, where the function $G$ represents either $S_{\text{NN}}$ or $E_{\text{NN}}$ and the matrix field $\boldsymbol{A}$ represents either $\boldsymbol{L}_{\text{NN}}$ or $\boldsymbol{M}_{\text{NN}}$. The most delicate part of this construction involves automatic satisfaction of the symmetry and degeneracy conditions in Eq. (3): the networks associated with the fields $\boldsymbol{A}$ are designed to ensure $\nabla_{\mathbf{z}}G \in \ker\boldsymbol{A}$ for the appropriate $G$. Inspired by ideas from matrix factorization, $\boldsymbol{A}$ is modeled by

$$\boldsymbol{A} := \boldsymbol{Q}_G^T \boldsymbol{B} \boldsymbol{Q}_G, \tag{4}$$

where $\boldsymbol{Q}_G$ is a matrix-valued function constructed based on skew-symmetric matrix fields $\boldsymbol{S}_j$, $j = 1, ..., K$, so that the $j$-th row of $\boldsymbol{Q}_G$ is defined as $(\boldsymbol{S}_j \nabla_{\mathbf{z}} G)^T$. This leads to the equality $\boldsymbol{Q}_G \nabla_{\mathbf{z}} G = \mathbf{0}$ and therefore $\boldsymbol{A}\nabla_{\mathbf{z}} G = \mathbf{0}$, i.e., the required degeneracy condition holds. Moreover, the matrix field $\boldsymbol{B}$ is skew-symmetric $(\boldsymbol{B}_L)$ if $\boldsymbol{A}$ represents $\boldsymbol{L}$ and symmetric positive semi-definite $(\boldsymbol{B}_M)$ if $\boldsymbol{A}$ represents $\boldsymbol{M}$, modeled by two triangular NNs,

$$\boldsymbol{B}_L := \boldsymbol{T}_L^T - \boldsymbol{T}_L \quad \text{and} \quad \boldsymbol{B}_M := \boldsymbol{T}_M^T \boldsymbol{T}_M. \tag{5}$$

To capture systems with parametric dependence, the networks associated with $G$ and $\boldsymbol{B}$ are parametrized by $\boldsymbol{\mu}$, allowing the proposed pGFINN to effectively capture parameter dependence while guaranteeing free energy conservation and entropy generation at each point of the parameter space. More detailed analysis on the effectiveness and performance of the proposed pGFINN is presented in Appendix A.4.

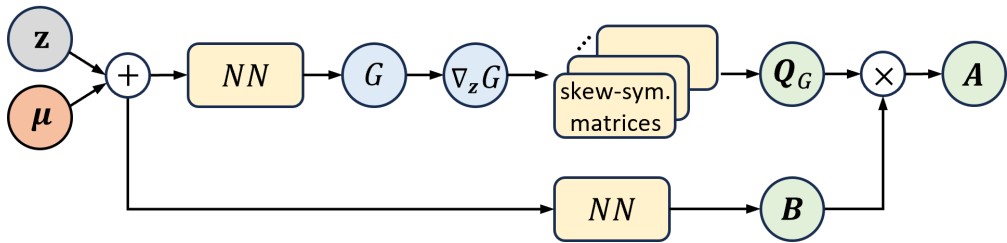

Figure 1: Architecture of the proposed pGFINN with trainable parameters highlighted in yellow.

## 2.2 Thermodynamics-Informed Latent Space Dynamics Identification (tLaSDI)

The tLaSDI presented in (Park et al., 2024) consists of a hyper-autoencoder for nonlinear dimensionality reduction combined with GFINN for learning thermodynamically consistent latent dynamics. The hyper-autoencoder in tLaSDI integrates a standard autoencoder (Hinton & Salakhutdinov, 2006) with a hyper-network (Ha et al., 2017), achieving a parametric latent representation of the system state through the generation of autoencoder coefficients based on a given input parameter $\boldsymbol{\mu}$. While this formulation does capture some form of parametric dependence in the governing equations, the hypernetwork-based parameterization substantially increases network complexity and training overhead, which may limit the scalability of tLaSDI for large-scale parametric dynamical systems.

To improve the efficiency and performance of parametric tLaSDI, we propose bypassing this hypernetwork entirely, instead integrating a standard autoencoder with a pGFINN, as described in Section 2.1. This has the advantage of incorporating parametric dependence directly into the learned dynamics, thereby maintaining consistency with the model from Eq. (1) and enabling an easier learning problem for the dimensionality reduction. Fig. 2 illustrates a schematic of the proposed tLaSDI in inference mode, after both the autoencoder and pGFINN have been trained. This inference proceeds as follows. Given a high-dimensional initial state $\mathbf{u}_0$ associated with the parameter $\boldsymbol{\mu}$, the encoder $\boldsymbol{\phi}_\mathrm{e}$ first transforms $\mathbf{u}_0$ to a low-dimensional latent state $\mathbf{z}_0$. The pGFINN $\boldsymbol{\psi}_\mathrm{pGFINN}$ then predicts thermodynamically consistent latent dynamics beginning at $\mathbf{z}_0$, which are subsequently projected back to the physical space by the decoder $\boldsymbol{\phi}_\mathrm{d}$.

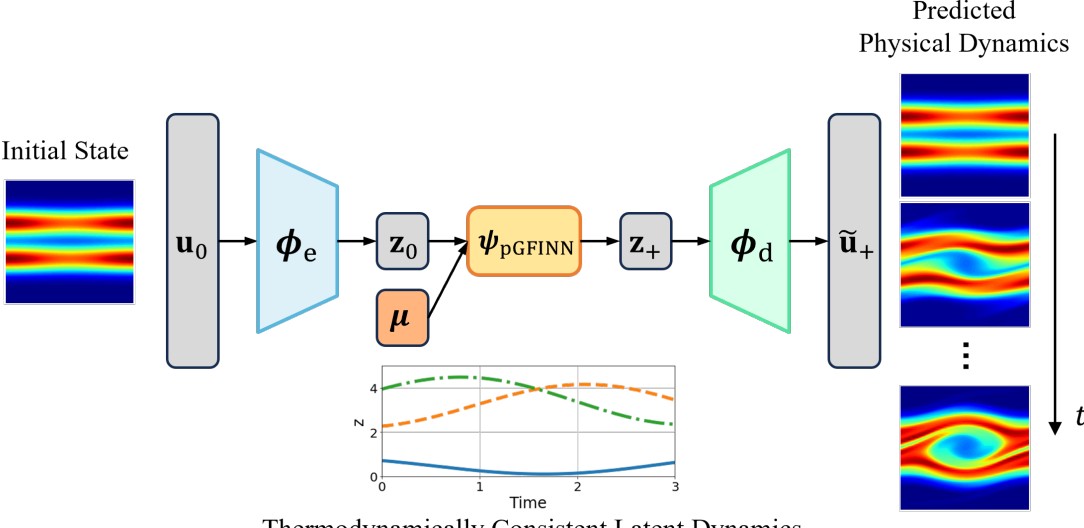

Figure 2: Schematic of tLaSDI for parametric dynamical systems in inference mode.

The loss function used to train these constituent parts is a sum of four terms:

$$\mathcal{L}(\boldsymbol{\theta}) = \mathcal{L}_\mathrm{int}(\boldsymbol{\theta}) + \lambda_\mathrm{rec}\mathcal{L}_\mathrm{rec}(\boldsymbol{\theta}) + \lambda_\mathrm{Jac}\mathcal{L}_\mathrm{Jac}(\boldsymbol{\theta}) + \lambda_\mathrm{mod}\mathcal{L}_\mathrm{mod}(\boldsymbol{\theta}), \tag{6}$$

where $\boldsymbol{\theta}$ is the collection of all trainable parameters. The first loss term $\mathcal{L}_\mathrm{int}$ represents the *integration* error,

$$\mathcal{L}_\mathrm{int} = \sum_{\boldsymbol{\mu}}\sum_{n} \left\| \boldsymbol{\phi}_\mathrm{e}(\mathbf{u}_{n+1}) - \boldsymbol{\phi}_\mathrm{e}(\mathbf{u}_n) - \int_{t_n}^{t_{n+1}} \boldsymbol{\psi}_\mathrm{pGFINN}\big(\boldsymbol{z}(t), \boldsymbol{\mu}, t\big)dt \right\|^2, \tag{7}$$

where the integral is approximated via a time integration scheme, e.g., a Runge–Kutta method. This term punishes discrepancy between the encoder-predicted latent state $\mathbf{z}_{n+1} = \phi_e(\mathbf{u}_{n+1})$ and the corresponding one-step integration of the latent dynamics starting from the previous encoder-predicted latent state $\mathbf{z}_n = \phi_e(\mathbf{u}_n)$.

Said differently, the integration term enforces the approximate equality

$$\phi_{\mathrm{e}}(\mathbf{u}_{n+1}) \approx \phi_{\mathrm{e}}(\mathbf{u}_n) + \int_{t_n}^{t_{n+1}} \psi_{\mathrm{pGFINN}}\big(\boldsymbol{z}(t), \boldsymbol{\mu}, t\big)dt, \tag{8}$$

which provides a consistency condition between pGFINN integration and the dynamics encoder.

The second loss term $\mathcal{L}_{\mathrm{rec}}$ represents the usual *reconstruction* error of the autoencoder, defined as

$$\mathcal{L}_{\mathrm{rec}} = \sum_{\boldsymbol{\mu}} \sum_n \left\| \mathbf{u}_n - \phi_{\mathrm{d}}\big(\phi_{\mathrm{e}}(\mathbf{u}_n)\big) \right\|^2, \tag{9}$$

which encourages the autoencoder to learn low-dimensional representations which effectively compress the solution data. The third loss term $\mathcal{L}_{\mathrm{Jac}}$ represents an infinitesimal regularity condition we call the *Jacobian* loss of the autoencoder,

$$\mathcal{L}_{\mathrm{Jac}} = \sum_{\boldsymbol{\mu}} \sum_n \left\| \big(\mathbf{I} - \mathbf{J}(\mathbf{u}_n)\big)\dot{\mathbf{u}}_n \right\|^2. \tag{10}$$

Here, $\mathbf{J}(\cdot) = \mathbf{J}_{\mathrm{d}}(\phi_{\mathrm{e}}(\cdot))\mathbf{J}_{\mathrm{e}}(\cdot)$ represents the Jacobian of the autoencoder, i.e., the derivative of the decoder's output with respect to the encoder's input and, $\mathbf{J}_{\mathrm{d}}(\cdot)$ resp. $\mathbf{J}_{\mathrm{e}}(\cdot)$ represent the Jacobian of the decoder resp. encoder. This is a heuristic, Sobolev-type loss that punishes infinitesimal inconsistency in the state $\mathbf{u}_n$ and its reconstruction after autoencoding, enforcing the condition

$$\dot{\mathbf{u}}_n \approx \frac{d}{dt}\phi_{\mathrm{d}}\big(\phi_{\mathrm{e}}(\mathbf{u}_n)\big) = \mathbf{J}_{\mathrm{d}}(\phi_{\mathrm{e}}(\mathbf{u}_n))\mathbf{J}_{\mathrm{e}}(\mathbf{u}_n)\dot{\mathbf{u}}_n = \mathbf{J}(\mathbf{u}_n)\dot{\mathbf{u}}_n. \tag{11}$$

In the case where the derivative data $\dot{\mathbf{u}}_n$ is unavailable, the Jacobian loss can be alternatively defined as

$$\mathcal{L}_{\mathrm{Jac}} = \sum_{\boldsymbol{\mu}} \sum_n \left\| \mathbf{I} - \mathbf{J}(\mathbf{u}_n) \right\|_F^2, \tag{12}$$

where $\|\cdot\|_F$ denotes the Frobenius norm. Finally, observe that the Jacobian of the autoencoder is rank-deficient since the latent dimension is much smaller than the physical dimension, implying that $\boldsymbol{J}(\cdot) \neq \boldsymbol{I}$ is never equal to the identity mapping. This fact leads to the final loss term $\mathcal{L}_{\mathrm{mod}}$, which accounts for the *modeling* error in tLaSDI. In particular, an addition of zero followed by the Cauchy-Schwarz inequality leads to the statement

$$\begin{aligned}
\big\|\big(\mathbf{I} - \mathbf{J}(\mathbf{u}_n)\big)\dot{\mathbf{u}}_n\big\| &\leq \big\|\dot{\mathbf{u}}_n - \mathbf{J}_{\mathrm{d}}(\phi_{\mathrm{e}}(\mathbf{u}_n))\psi_{\mathrm{pGFINN}}\big(\phi_{\mathrm{e}}(\mathbf{u}_n)\big)\big\| \\
&\quad + \big\|\mathbf{J}_{\mathrm{d}}(\phi_{\mathrm{e}}(\mathbf{u}_n))\big\| \cdot \big\|\mathbf{J}_{\mathrm{e}}(\mathbf{u}_n)\dot{\mathbf{u}}_n - \psi_{\mathrm{pGFINN}}\big(\phi_{\mathrm{e}}(\mathbf{u}_n)\big)\big\|,
\end{aligned} \tag{13}$$

where the first term on the right-hand side quantifies the modeling error of the full-order dynamics, obtained by projecting the latent dynamics predicted by pGFINN through the decoder. Conversely, the second term measures the modeling error between the encoder-derived representation of the velocity field, $\mathbf{J}_{\mathrm{e}}(\mathbf{u}_n)\dot{\mathbf{u}}_n$, and the pGFINN prediction, $\psi_{\mathrm{pGFINN}}\big(\phi_{\mathrm{e}}(\mathbf{u}_n)\big)$. Accordingly, $\mathcal{L}_{\mathrm{mod}}$ is defined as

$$\mathcal{L}_{\mathrm{mod}} = \sum_{\boldsymbol{\mu}} \sum_n \left\| \dot{\mathbf{u}}_n - \mathbf{J}_{\mathrm{d}}(\phi_{\mathrm{e}}(\mathbf{u}_n))\psi_{\mathrm{pGFINN}}\big(\phi_{\mathrm{e}}(\mathbf{u}_n)\big) \right\|^2 + \left\| \mathbf{J}_{\mathrm{e}}(\mathbf{u}_n)\dot{\mathbf{u}}_n - \psi_{\mathrm{pGFINN}}\big(\phi_{\mathrm{e}}(\mathbf{u}_n)\big) \right\|^2. \tag{14}$$

The comprehensive loss function Eq. (6) reflects an error bound for tLaSDI under the assumptions that the decoder's Jacobian is bounded and Lipschitz continuous, and that the pGFINN predictions are bounded (Park et al., 2024). Advantageously, it enables simultaneous training of the autoencoder and the latent space dynamics model, ensuring consistency in the learned latent dynamics while minimizing modeling errors in the full-order system. The advantages of such joint training have been previously demonstrated in the literature (Champion et al., 2019; He et al., 2023; Bonneville et al., 2024a; Vijayarangan et al., 2024; Park et al., 2024; He et al., 2025), highlighting that enforcing interactions between the autoencoder and dynamics models, and thereby imposing structure on the latent dynamics, is crucial for improving model performance and generalization. An upper bound on the tLaSDI state error given in terms of the loss function Eq. (6)

is presented in Appendix A.1, offering a theoretical foundation for this training approach. More detailed discussion about the effects of the Jacobian and modeling loss terms on model performance can be found in the Appendix A.2.

Note that constructing the full tLaSDI loss in Eq. (6) requires derivative information as training data, especially for the Jacobian loss in Eq. (10) and the modeling loss in Eq. (14). In the case that this is prohibitive (such as in the presence of noisy conditions), the strong form ODE in Eq. (1) can be transformed to a weak form by multiplying both sides with a continuous, compactly supported test function and integrating over its support. By applying integration by parts and exploiting the compact support of the test function, temporal derivatives of the state variables are transferred to the test function, thereby stabilizing derivative computation in the presence of noisy data and eliminating the need for pointwise derivatives of the input (Messenger & Bortz, 2021a). Recent studies have successfully incorporated this weak formulation into latent-space dynamics identification frameworks to improve the robustness and accuracy of ROM in the presence of noise (Tran et al., 2024; He et al., 2025; Bonneville et al., 2024b). The proposed tLaSDI method can be extended in a similar manner, although this is beyond the scope of the present study.

### 2.3 Physics-Informed Active Learning

Another important component of tLaSDI training relies on adaptive sampling. To effectively explore the parameter space and enhance model performance, tLaSDI adopts a greedy, physics-informed active learning strategy (He et al., 2023) which adaptively samples informative training data on the fly. Given a testing parameter $\boldsymbol{\mu}^*$, modeling accuracy is assessed using a physics-informed error indicator derived from the residual Eq. (2) of the governing equations Eq. (1):

$$e_{res} = \sum_n ||\mathbf{r}(\tilde{\mathbf{u}}_n; \tilde{\mathbf{u}}_{n-1}, \boldsymbol{\mu}^*)||, \tag{15}$$

where $\tilde{\mathbf{u}}$ represents approximate full-order dynamics through both the autoencoder and the latent dynamics learned by pGFINN. Typically, the sum on time steps $n$ is sparsified, since predictions at a small subset of time steps suffice to obtain a reasonable measurement of the error. This residual-based error indicator depends solely on model predictions and shows a strong positive correlation with the maximum relative error, enabling efficient and effective adaptation across parameter space. When adaptive sampling is enabled, tLaSDI training begins with a limited number of parameter points (training samples), often located at the corners of the parameter space. Every $N_{up}$ epochs, the tLaSDI model is evaluated at a random set of testing parameters, computing the error indicator defined in Eq. (15) at each parameter value. The parameter point corresponding to the maximum error is then added to the training set before training resumes. This physics-informed active learning process continues until a predefined number of training samples is reached or the desired model accuracy is achieved. Note that the mild computational overhead incurred by this procedure is generally outweighed by its performance improvement over simple uniform sampling (c.f. Section 3.1), leading to a simple and robust boost to the generalization capabilities of the proposed tLaSDI model.

## 3 Experiments

We now demonstrate the performance of the proposed tLaSDI framework by applying it to common benchmarks in nonlinear model reduction: the 1D Burgers' equation and the 1D/1V Vlasov two-stream plasma instability problem.

### 3.1 One-dimensional Burgers' Equation

We begin by considering a one-dimensional (1D) inviscid Burgers' equation with periodic boundary conditions and a Gaussian initial condition parameterized by $\boldsymbol{\mu} = \{a, w\} \in \mathcal{D}$,

$$
\begin{cases}
\dfrac{\partial u}{\partial t} + u\dfrac{\partial u}{\partial x} = 0, & t \in [0, 1], \quad x \in [-3, 3] \\
u(t, x = 3) = u(t, x = -3) \\
u(t = 0, x; \boldsymbol{\mu}) = a \exp\left(-\dfrac{x^2}{2w^2}\right).
\end{cases}
\tag{16}
$$

The parameter space is defined as $\mathcal{D} = [0.7, 0.9] \times [0.9, 1.1]$, associated with the magnitude and width of the initial Gaussian pulse. The implementation details can be found in Appendix A.5. The hypernetwork-based tLaSDI and the proposed pGFINN-based tLaSDI are trained with 9 parameter points uniformly distributed in the parameter space $\mathcal{D}$. Fig. 3 presents the maximum relative errors of the trained models across this parameter space, where Figs. 3(a-b) show that pGFINN-based tLaSDI outperforms the hyper-autoencoder-based approach, highlighting the effectiveness of the proposed pGFINN parameterization. Moreover, the training cost of pGFINN-based tLaSDI is reduced by 50% over the previous hypernetwork variant, while the inference cost is lowered by 61%, demonstrating that the proposed approach also provides significant reductions in computational burden.

To demonstrate the effectiveness of the physics-informed active learning strategy described in Section 2.3, we train an additional pGFINN-based tLaSDI integrated with this strategy. The training begins with 4 parameter points located at the corners of the parameter space and adaptively selects 4 additional parameter points using the active learning approach, resulting in a total of 8 training points by the end of training. Although the total number of training points is less than that used in the case of uniform sampling, Figs. 3(b-c) demonstrate that the performance of pGFINN-based tLaSDI is further enhanced by the proposed active learning strategy, without increasing the cost of model training. Notably, active learning adaptively selects additional parameter points in regions of higher modeling error, particularly near the lower end of the parameter space, as shown in Fig. 3(b). This leads to enhanced model accuracy across the entire range of parameters considered, significantly outperforming the case of uniform sampling especially in regions where errors are highest.

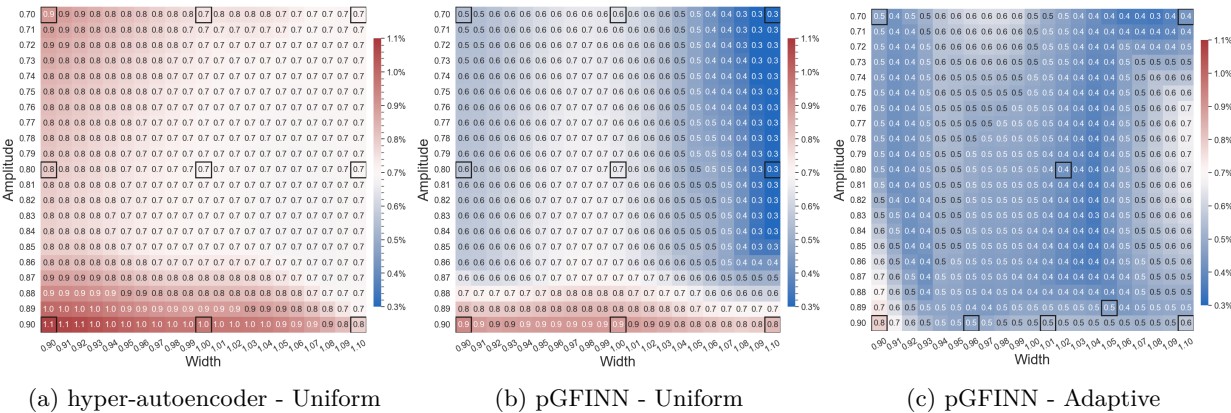

(a) hyper-autoencoder - Uniform     (b) pGFINN - Uniform     (c) pGFINN - Adaptive

Figure 3: Comparison of the maximum relative errors across the parameter space from (a) hypernetwork-based tLaSDI with uniform sampling, (b) pGFINN-based tLaSDI with uniform sampling, and (c) pGFINN-based tLaSDI with adaptive sampling. The training points are highlighted with black boxes.

### 3.2 1D/1V Vlasov–Poisson Equation

The effectiveness of the proposed tLaSDI framework is further examined using the 1D/1V Vlasov–Poisson equation with initial conditions parameterized by the temperature $T$ and the wavenumber $k$, i.e., $\boldsymbol{\mu} =$

$\{T, k\} \in \mathcal{D} = [0.9, 1.1] \times [1.0, 1.2]$. These equations are given by

$$\begin{cases} \dfrac{\partial f}{\partial t} + \dfrac{\partial v f}{\partial x} + \dfrac{\partial}{\partial v}\left(\dfrac{d\Phi}{dx}f\right) = 0, \quad t \in [0,5], \quad x \in [0,2\pi] \quad v \in [-7,7] \\ \dfrac{d^2\Phi}{dx^2} = \displaystyle\int_v f \, dv \\ f(t=0, x, v; \boldsymbol{\mu}) = \dfrac{8}{\sqrt{2\pi T}}\left[1 + \dfrac{1}{10}\cos(kx)\right]\left[\exp\left(-\dfrac{(v-2)^2}{2T}\right) + \exp\left(-\dfrac{(v+2)^2}{2T}\right)\right], \end{cases} \quad (17)$$

where $f(x, v)$ is the plasma distribution function, dependent on a spatial coordinate $x$ and a velocity co-ordinate $v$, and $\Phi$ is the electrostatic potential. This model describes the dynamics of a one-dimensional, collisionless, electrostatic plasma, and is representative of the complex models for plasma behaviors observed in nuclear fusion reactors. Owing to dependence on the velocity variable, it is a two-dimensional PDE despite its single spatial dimension. The physical dynamics corresponding to the parameter case $(T = 0.9, k = 1.0)$ are illustrated in Fig. 4.

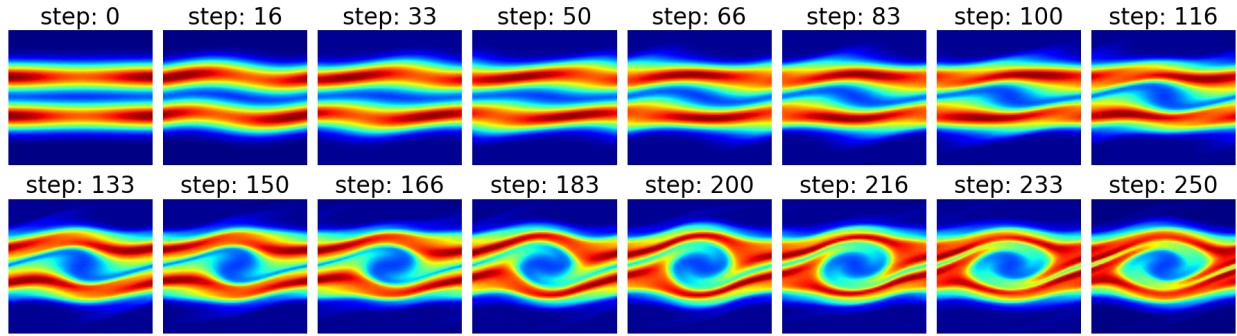

Figure 4: 1D/1V Vlasov - The physical dynamics of the parameter case $(T = 0.9, k = 1.0)$.

In this more challenging example, the training data include 16 parameter points, uniformly distributed across the parameter space $\mathcal{D}$. The implementation details can be found in Appendix A.5. Figs. 5(a-b) compare the latent dynamics learned by hypernetwork-based tLaSDI and the proposed pGFINN-based tLaSDI. Observe that the proposed tLaSDI model learns simpler latent dynamics and exhibits stronger consistency between the latent state encoding and the pGFINN dynamics prediction. This is further confirmed by examining the frequency-domain representation of the latent dynamics, shown in Figs. 5(c-d), which reveals that the frequency content of the latent dynamics generated by the proposed tLaSDI concentrate at lower Hertz values than those in the model with hyper-autoencoder, further highlighting the simplicity of the learned dynamics.

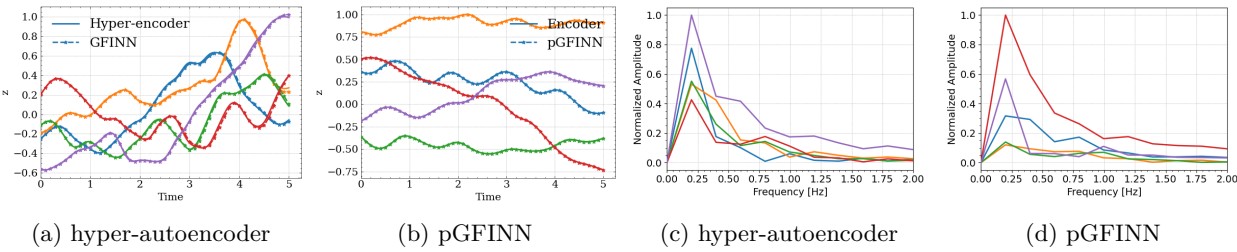

| (a) hyper-autoencoder | (b) pGFINN | (c) hyper-autoencoder | (d) pGFINN |

Figure 5: 1D/1V Vlasov - Comparison of the latent dynamics learned by (a) hypernetwork-based tLaSDI and (b) pGFINN-based tLaSDI; the corresponding frequencies for (c) hypernetwork-based tLaSDI and (d) pGFINN-based tLaSDI.

Figs. 6 demonstrate that the simpler and more consistent latent dynamics of the proposed tLaSDI also contribute to higher model accuracy, achieving an average error of 1.66% across the entire parameter space,

compared to 2.18% for its hyper-autoencoded counterpart, a 24% reduction in error with remarkably less training complexity. The significantly lower errors observed at the training parameter points for the proposed tLaSDI suggest that the pGFINN which propagates latent dynamics more effectively captures parameterization in the governing Eq. (1), as opposed to the hyper-autoencoded parameter-dependence alongside static latent dynamics. Additionally, due to the comparatively smaller size of the pGFINN-based tLaSDI, model accuracy is expected to improve as the training dataset expands with less risk of overfitting. Furthermore, the training cost of the proposed tLaSDI is reduced by 90%, while the inference cost is lowered by 57% – indicating a highly impactful reduction in computational burden. Compared with the high-fidelity simulation code (HyPar (Ghosh, 2025)) running a spatially coarsened configuration that produces a similar error (1.66%) with respect to the training data, the proposed tLaSDI model achieves a $2,495\times$ speed-up. The time step size used to generate the training data is $5 \times 10^{-3}$. Increasing the time step size of the high-fidelity simulations to $2 \times 10^{-2}$ while keeping the spatial resolution fixed reduces the computational time and incurs small relative errors of 0.014% with respect to the training data computed at the smaller time step size. Further increasing the time step size leads to unstable simulations and convergence issues. When compared to the high-fidelity simulation with the largest time step size of $2 \times 10^{-2}$, the proposed tLaSDI model achieves a 1,370$\times$ speed-up. Although the examples in this paper rely on simulated data generated from PDE solvers, we emphasize that the proposed tLaSDI model is equally applicable to settings involving dynamic sensor data where the governing ODEs/PDEs are not known a priori. More detailed discussion about the speed-up analysis can be found in the Appendix A.3.

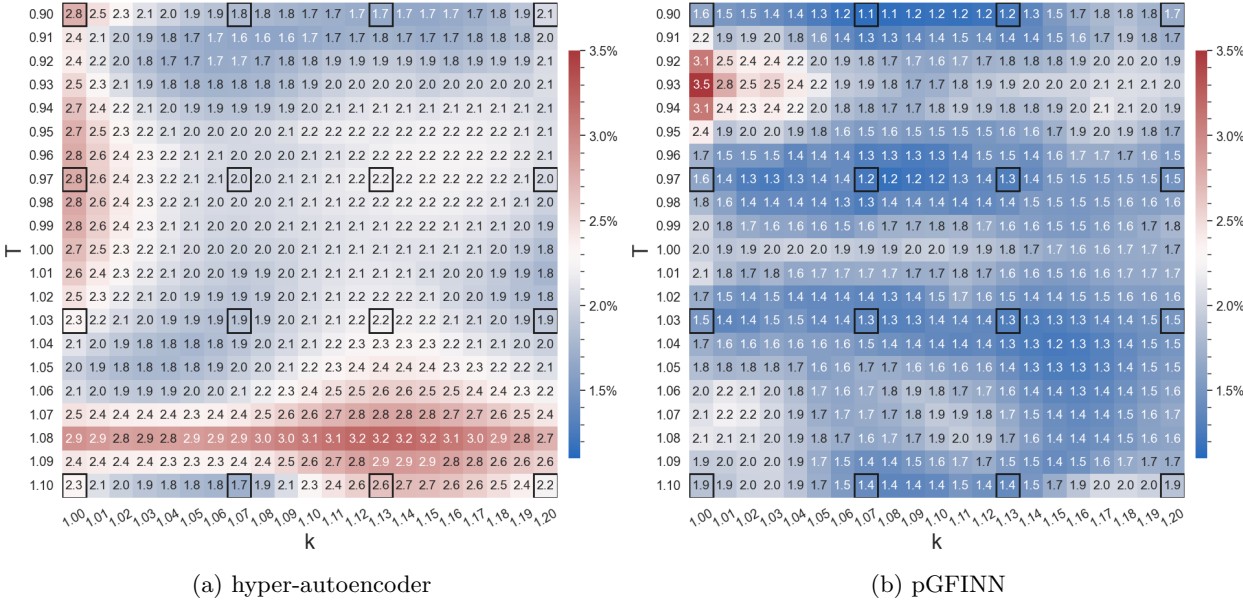

(a) hyper-autoencoder             (b) pGFINN

Figure 6: 1D/1V Vlasov - Comparison of the maximum relative errors across the parameter space between (a) hypernetwork-based tLaSDI and (b) pGFINN-based tLaSDI. The training points are highlighted with black boxes.

To confirm the structure-preservation properties of the proposed approach which are guaranteed by the pGFINN component, Fig. 7 illustrates a roll out of the learned energy and entropy in the latent space and their rates for various parameter instances. As expected, the energy conservation and increasing entropy production, demonstrated in Fig. 7, confirm that the learned latent dynamics satisfy the first two laws of thermodynamics as guaranteed by the GENERIC formalism. Moreover, these dynamics are remarkably interpretable as a consequence of GENERIC structure-preservation. Observe from Fig. 7(d) that, for fixed temperature $T$, increasing the wavenumber $k$ leads to higher initial entropy and lower final entropy. Similarly, Fig. 7(e) shows that, for fixed wavenumber $k$, increasing temperature $T$ leads to a higher initial entropy and lower final entropy. These trends are consistent with known plasma physics in the high-dimensional space, since increasing temperature and wavenumber suppresses growth in the two-stream instability (Koide et al., 2023). This demonstrates the ability of the proposed tLaSDI to capture physically meaningful trends in the

high-fidelity system independently of its accuracy, since each realization is guaranteed to exhibit meaningful thermodynamics. Additionally, the entropy production rates shown in Fig. 4(f) exhibit peaks at around steps 15, 50, 75, 120, and 180, which correspond meaningfully to the growth rate of the two stream instability. These results suggest that the thermodynamic behavior within the latent space may provide valuable insights into the dynamics occurring in the physical space.

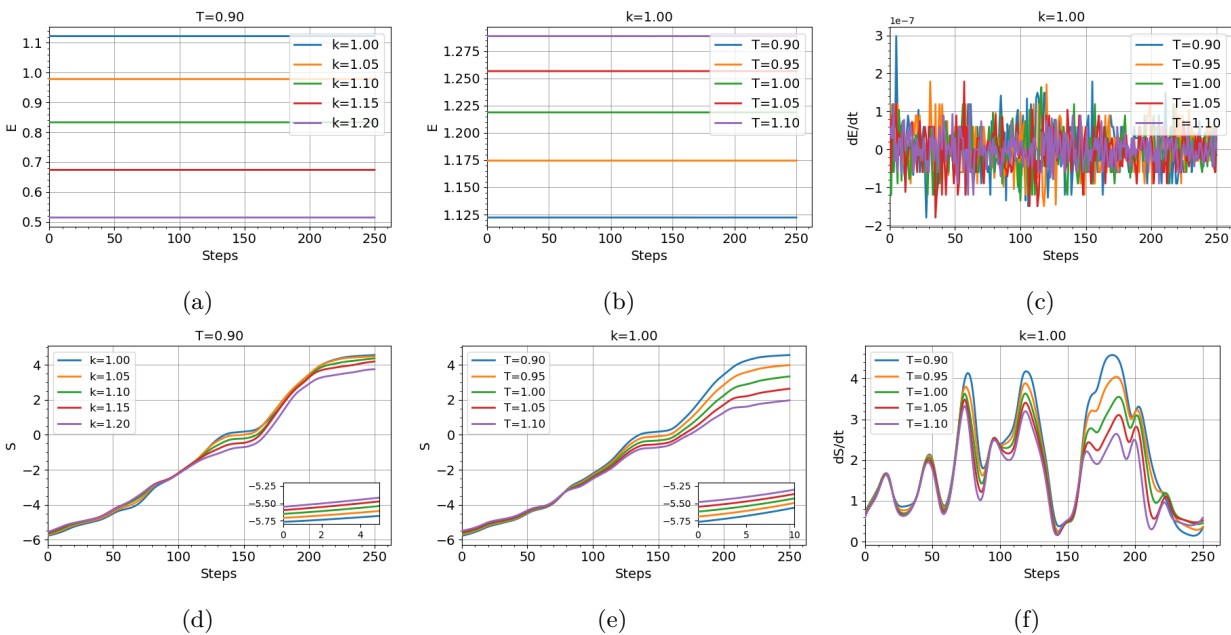

Figure 7: 1D/1V Vlasov - The evolution of energy for: (a) $T = 0.9$ and varying $k$; (b) $k = 1.0$ and varying $T$; the evolution of energy rates for (c) $k = 1.0$ and varying $T$; the evolution of entropy for: (c) $T = 0.9$ and varying $k$; (d) $k = 1.0$ and varying $T$; the evolution of entropy production rates for (e) $k = 1.0$ and varying $T$.

## 3.3 Baseline Comparison

In this section, we benchmark the proposed tLaSDI framework against two state-of-the-art operator learning methods, including Deep Operator Networks (DeepONets) (Lu et al., 2021; 2022) and the Fourier Neural Operator (FNO) (Li et al., 2020). The test problem considered is the 1D/1V Vlasov-Poisson equation (see Section 3.2).

For this experiment, a three-dimensional FNO (FNO-3D) is used that directly performs convolution in space and time. Both DeepONet and FNO-3D are trained using the same dataset, described in Section 3.2, with implementation details provided in Appendix A.5. All models take the full-order initial state as input and predict the solution at future time steps. To ensure a fair comparison, each model is configured to maintain a comparable architecture and number of trainable parameters to our proposed pGFINN-based tLaSDI (c.f. Appendix A.5.2).

Fig. 8 shows the maximum relative errors across the parameter space for DeepONet and FNO-3D, which achieve average errors of 2.87% and 2.20%, respectively. Notably, DeepONet exhibits larger errors across most regions of the parameter space, whereas FNO-3D demonstrates lower errors at training points compared to nearby testing points. This pattern is also observed in the proposed tLaSDI, as shown in Fig. 6(b). Table 1 compares the performance of the tLaSDI models against DeepONet and FNO-3D models. Remarkably, the proposed tLaSDI achieves superior accuracy across the parameter space in this example, with an average error of 1.66%.

While accuracy is certainly a useful metric by which to gauge model performance, it is important to note some additional advantages of tLaSDI which are unique to its construction. First, it can be seen that both

Table 1: Comparison of performance on 1D/1V Vlasov-Poisson equation

| Model | Parameters | Time (s) / Epoch | Mean Error (%) |
|---|---|---|---|
| tLaSDI (pGFINN) | 480,367 | 0.053 | **1.663** |
| tLaSDI (Hyper-Autoencoder) | 9,526,427 | 0.527 | 2.176 |
| DeepONet | 481,361 | 0.042 | 2.870 |
| FNO-3D | 486,336 | 0.687 | 2.200 |

DeepONet and FNO require parameterization through the initial condition, i.e., they learn a direct mapping from the initial state to future states after (implicit) temporal evolution. However, in applications where parameters affect aspects of the dynamics other than the initial condition, such as the system properties $\alpha, m, k$ seen in previous experiments, the full-order initial state remains identical across the entire parameter space. In such scenarios, DeepONet and FNO cannot directly account for parameter dependence, thereby limiting both their performance and applicability. In contrast, the proposed parametric tLaSDI can naturally accommodate arbitrary forms of parameterization by decoupling parameter dependence from the initial state. Furthermore, DeepONet and FNO are presently limited to pre-defined sampling strategies, while tLaSDI leverages physics-informed active learning for adaptive sampling, enhancing both its efficiency and predictive performance in generalization tasks.

Perhaps most importantly, unlike the black-box nature of DeepONet and FNO, the proposed framework strongly enforces thermodynamics principles to enable interpretable, structure-preserving learning. This enforcement not only guarantees important mathematical properties of the learned dynamics such as global asymptotic stability, but also enhances generalization robustness and provides deeper insights into the underlying physical dynamics, without compromising predictive accuracy, as demonstrated in Section 3.2. The benchmark results seen here highlight the potential of tLaSDI as an efficient, interpretable, and physically consistent reduced-order modeling approach for parametric dynamical systems.

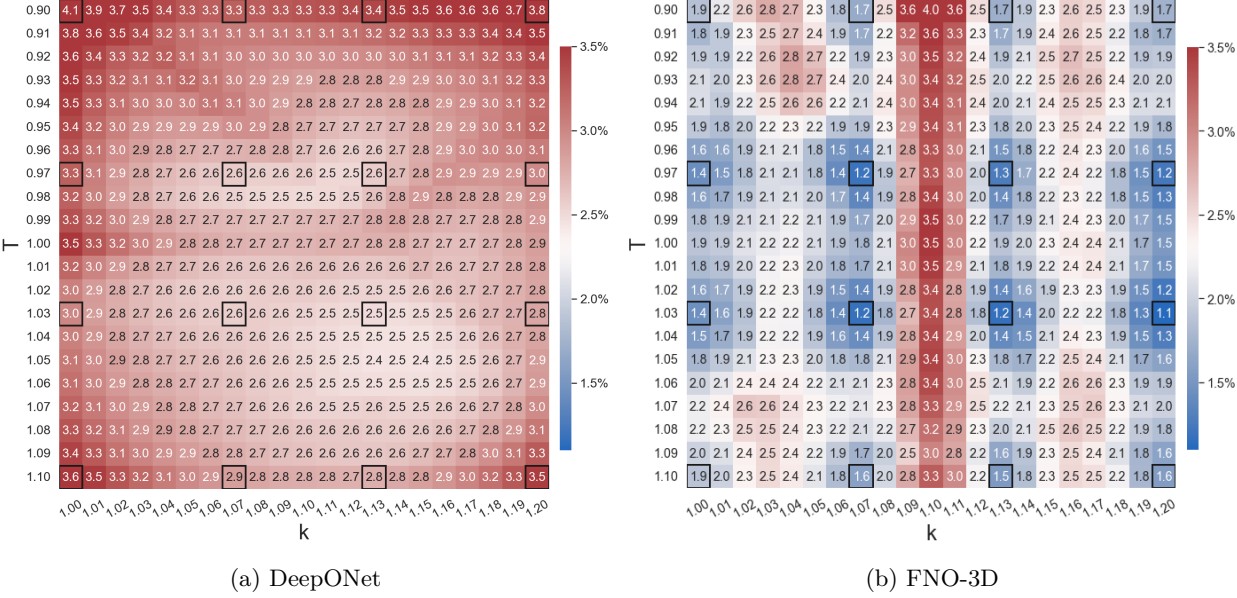

(a) DeepONet

(b) FNO-3D

Figure 8: 1D/1V Vlasov - Maximum relative errors across the parameter space: (a) DeepONet (b) FNO-3D. The training points are highlighted with black boxes.

**Limitations**

This work has restricted investigation to problems with moderate parameter dimensionality and physical-state dimension. Although the proposed approach can, in principle, be extended to high-dimensional parameter spaces and large-scale physical states, doing so would primarily serve to assess scalability and computational cost rather than the core methodological contributions. Accordingly, the experiments presented here should be regarded as proof-of-concept benchmarks that validate the fundamental properties of the pGFINN+tLaSDI framework, leaving a systematic study of scalability to future work.

## 4 Conclusion

We have presented an enhanced, thermodynamics-informed latent space dynamics identification (tLaSDI) framework for efficient, interpretable, and physically consistent reduced-order modeling of parametric dynamical systems. The newly developed parametric GENERIC formalism-informed neural networks (pGFINNs), designed for use within the proposed tLaSDI, guarantees conservation of free energy and generation of entropy in the latent space across variations in system parameters, ensuring thermodynamically consistent latent dynamics for any parametric realization. In addition, the integration of a physics-informed active learning strategy further improves model performance by adaptively selecting informative parameter samples based on a residual-based error indicator, leading to a tLaSDI method exhibiting simple latent dynamics along with highly substantial improvements over previous methods on standard benchmarks. In particular, the proposed approach achieves a 50%-90% reduction in training cost and 57%-61% reduction in inference time over tLaSDI models incorporating parametric dependence through a hyper-autoencoder. Compared to the high-fidelity simulation, it also achieves up to a $2,495\times$ speed-up with 1-3% relative errors. Perhaps most importantly, the learned latent dynamics reveal interpretable thermodynamic behavior, offering valuable insights into the physical system's evolution which can be used to guide control and analysis. Ultimately, these results highlight the potential of the proposed tLaSDI framework as a scalable, data-efficient, and physically grounded foundation for the next generation of reduced-order models in complex, high-dimensional parametric settings.

Although the experiments in this study focus on parameterization through initial conditions, the proposed framework is readily extensible to other forms of parameterization, such as material properties and constitutive relationships, which are particularly relevant for inverse problems. While the use of a physics-informed error indicator for active learning relies on access to the high-fidelity solver, which may not be feasible in data-only scenarios or certain practical applications, the framework can be adapted by incorporating a Gaussian process-based active learning strategy for adaptive sampling (Bonneville et al., 2024a). Since derivative computations may become oscillatory in scenarios involving noisy data, thereby affecting latent dynamics identification and model performance, a weak-form formulation can be leveraged to reduce variance and achieve robust latent space dynamics identification under noisy conditions (Messenger & Bortz, 2021a; Tran et al., 2024; He et al., 2025; Bonneville et al., 2024b). Moreover, the proposed tLaSDI framework is general and not limited to the use of standard autoencoders; alternative dimensionality reduction techniques, such as convolutional autoencoders or linear compression techniques like POD, may be employed to improve training efficiency. Future work will consider the integration of an automatic neural architecture search (Pham et al., 2018; Ren et al., 2021) to further optimize the autoencoder design for enhanced generalization performance.

**Acknowledgments**

This work was supported by the U.S. Department of Energy (DOE), Office of Science, Office of Advanced Scientific Computing Research (ASCR), through the CHaRMNET Mathematical Multifaceted Integrated Capability Center (MMICC) under Award Number DE-SC0023164 to YC, and through the SEA-CROGS MMICC under Award Number DE-SC0023191 supporting AG. Livermore National Laboratory is operated by Lawrence Livermore National Security, LLC, for the U.S. Department of Energy, National Nuclear Security Administration under Contract DE-AC52-07NA27344. LLNL document release number: LLNL-CONF-2006917. This article has been co-authored by an employee of National Technology & Engineering Solutions of Sandia, LLC under Contract No. DE-NA0003525 with the U.S. Department of Energy (DOE). The employee owns all right, title and interest in and to the article and is solely responsible for its contents. The United

States Government retains and the publisher, by accepting the article for publication, acknowledges that the United States Government retains a non-exclusive, paid-up, irrevocable, world-wide license to publish or reproduce the published form of this article or allow others to do so, for United States Government purposes. The DOE will provide public access to these results of federally sponsored research in accordance with the DOE Public Access Plan `https://www.energy.gov/downloads/doe-public-access-plan`. This paper describes objective technical results and analysis. Any subjective views or opinions that might be expressed in the paper do not necessarily represent the views of the U.S. Department of Energy or the United States Government. KL acknowledges support from the U.S. National Science Foundation under grant IIS 2338909. YS was partially supported by the National Science Foundation under the grant DMS-2513966 (Computational Mathematics Program) and the NRF grant funded by the Ministry of Science and ICT of Korea (RS-2023-00219980).

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

# A    Appendix

## A.1    Error Analysis

This section provides a bound on the error of the tLaSDI approximation similar to Park et al. (2024), along with a proof for completeness. Note that parametric dependence will be largely ignored in the following arguments; the resulting bound can be assumed to hold pointwise for each value of the parameters $\boldsymbol{\mu}$. First, observe that the error in the tLaSDI approximation is expressible as

$$\mathbf{e}(t) = \mathbf{u}(t) - \tilde{\mathbf{u}}(t) = \big(\mathbf{u}(t) - \hat{\mathbf{u}}(t)\big) + \big(\hat{\mathbf{u}}(t) - \tilde{\mathbf{u}}(t)\big), \tag{18}$$

where $\hat{\mathbf{u}}(t) = \boldsymbol{\phi}_{\mathrm{d}}\big(\boldsymbol{\phi}_{\mathrm{e}}(\mathbf{u}(t))\big)$ represents the reconstructed full-order dynamics from the autoencoder and $\tilde{\mathbf{u}}(t) = \boldsymbol{\phi}_{\mathrm{d}}\big(\mathbf{z}(t)\big)$ denotes the tLaSDI approximation. Assuming the initial latent state $\mathbf{z}(t_0) = \boldsymbol{\phi}_{\mathrm{e}}\big(\mathbf{u}(t_0)\big)$, the latent tLaSDI dynamics are given by $\dot{\mathbf{z}} = \mathbf{f}^r(\mathbf{z})$ where $\mathbf{f}^r$ denotes the pGFINN. The goal is to bound the method error $\mathbf{e}(t)$ in terms of quantities related to the loss terms $\mathcal{L}_{\mathrm{int}}, \mathcal{L}_{\mathrm{rec}}, \mathcal{L}_{\mathrm{Jac}}$, and $\mathcal{L}_{\mathrm{mod}}$.

To this end, observe that $\mathbf{e}_{\mathrm{AE}}(t) = \mathbf{u}(t) - \hat{\mathbf{u}}(t)$ accounts for the reconstruction error of the autoencoder, whereas $\mathbf{e}_{\mathrm{DI}}(t) = \hat{\mathbf{u}}(t) - \tilde{\mathbf{u}}(t)$ accounts for the error of latent space dynamics identification. It follows that there is an initial value problem for the reconstruction error,

$$\dot{\mathbf{e}}_{\mathrm{AE}} = \dot{\mathbf{u}} - \dot{\hat{\mathbf{u}}} = (\mathbf{I} - \mathbf{J}(\mathbf{u}))\dot{\mathbf{u}}, \qquad \mathbf{e}_{\mathrm{AE}}(t_0) = \mathbf{u}(t_0) - \boldsymbol{\phi}_{\mathrm{d}}\big(\boldsymbol{\phi}_{\mathrm{e}}(\mathbf{u}(t_0))\big). \tag{19}$$

Integrating this over time produces the solution

$$\mathbf{e}_{\mathrm{AE}} = \mathbf{e}_{\mathrm{AE}}(t_0) + \int_{t_0}^{t} \big(\mathbf{I} - \mathbf{J}(\mathbf{u}(s))\big)\dot{\mathbf{u}}(s)ds, \tag{20}$$

which can be bounded for any $\alpha \in [0, 1]$,

$$\|\mathbf{e}_{\mathrm{AE}}(t)\| \leq \alpha\Big(\|\mathbf{e}_{\mathrm{AE}}(t_0)\| + \int_{t_0}^{t} \big\|\big(\mathbf{I} - \mathbf{J}(\mathbf{u}(s))\big)\dot{\mathbf{u}}(s)\big\|ds\Big) + (1 - \alpha)\|\mathbf{e}_{\mathrm{AE}}(t)\|. \tag{21}$$

Turning attention to the latent dynamics identification, there is a corresponding initial value problem,

$$\dot{\mathbf{e}}_{\mathrm{DI}} = \dot{\hat{\mathbf{u}}} - \dot{\tilde{\mathbf{u}}} = \mathbf{J}(\mathbf{u})\mathbf{f}(\mathbf{u}) - \mathbf{J}_{\mathrm{d}}(\mathbf{z})\mathbf{f}^r(\mathbf{z}), \qquad \mathbf{e}_{\mathrm{DI}}(t_0) = \mathbf{0}. \tag{22}$$

Observe that an addition of zero on the right-hand side leads to

$$\mathbf{J}(\mathbf{u})\mathbf{f}(\mathbf{u}) - \mathbf{J}_{\mathrm{d}}(\mathbf{z})\mathbf{f}^r(\mathbf{z}) = \Big(\mathbf{J}(\mathbf{u})\mathbf{f}(\mathbf{u}) - \mathbf{J}_{\mathrm{d}}(\boldsymbol{\phi}_{\mathrm{e}}(\mathbf{u}))\mathbf{f}^r(\mathbf{z})\Big) + \Big(\mathbf{J}_{\mathrm{d}}(\boldsymbol{\phi}_{\mathrm{e}}(\mathbf{u}))\mathbf{f}^r(\mathbf{z}) - \mathbf{J}_{\mathrm{d}}(\mathbf{z})\mathbf{f}^r(\mathbf{z})\Big), \tag{23}$$

which can be bounded with the triangle inequality to yield

$$\begin{aligned}\big\|\mathbf{J}(\mathbf{u})\mathbf{f}(\mathbf{u}) - \mathbf{J}_{\mathrm{d}}(\mathbf{z})\mathbf{f}^r(\mathbf{z})\big\| &\leq \big\|\mathbf{J}_{\mathrm{d}}(\boldsymbol{\phi}_{\mathrm{e}}(\mathbf{u}))\big\| \cdot \big\|\mathbf{J}_{\mathrm{e}}(\mathbf{u})\mathbf{f}(\mathbf{u}) - \mathbf{f}^r(\mathbf{z})\big\| \\ &\quad + \big\|\mathbf{J}_{\mathrm{d}}(\boldsymbol{\phi}_{\mathrm{e}}(\mathbf{u})) - \mathbf{J}_{\mathrm{d}}(\mathbf{z})\big\| \cdot \big\|\mathbf{f}^r(\mathbf{z})\big\| \end{aligned} \tag{24}$$

On the other hand, a different addition of zero gives the alternative expression

$$\mathbf{J}(\mathbf{u})\mathbf{f}(\mathbf{u}) - \mathbf{J}_{\mathrm{d}}(\mathbf{z})\mathbf{f}^r(\mathbf{z}) = \Big(\mathbf{J}(\mathbf{u})\mathbf{f}(\mathbf{u}) - \mathbf{f}(\mathbf{u})\Big) + \Big(\mathbf{f}(\mathbf{u}) - \mathbf{J}_{\mathrm{d}}(\mathbf{z})\mathbf{f}^r(\mathbf{z})\Big), \tag{25}$$

which is similarly bounded as

$$\big\|\mathbf{J}(\mathbf{u})\mathbf{f}(\mathbf{u}) - \mathbf{J}_{\mathrm{d}}(\mathbf{z})\mathbf{f}^r(\mathbf{z})\big\| \leq \big\|(\mathbf{I} - \mathbf{J}(\mathbf{u}))\mathbf{f}(\mathbf{u})\big\| + \big\|\mathbf{f}(\mathbf{u}) - \mathbf{J}_{\mathrm{d}}(\mathbf{z})\mathbf{f}^r(\mathbf{z})\big\|. \tag{26}$$

Therefore, an $\alpha$-linear combination of Eqs. (24) and (26) yields

$$\begin{aligned}\big\|\mathbf{J}(\mathbf{u})\mathbf{f}(\mathbf{u}) - \mathbf{J}_{\mathrm{d}}(\mathbf{z})\mathbf{f}^r(\mathbf{z})\big\| &\leq \alpha\big\|\mathbf{J}_{\mathrm{d}}(\boldsymbol{\phi}_{\mathrm{e}}(\mathbf{u}))\big\| \cdot \big\|\mathbf{J}_{\mathrm{e}}(\mathbf{u})\mathbf{f}(\mathbf{u}) - \mathbf{f}^r(\mathbf{z})\big\| \\ &\quad + \alpha\big\|\mathbf{J}_{\mathrm{d}}(\boldsymbol{\phi}_{\mathrm{e}}(\mathbf{u})) - \mathbf{J}_{\mathrm{d}}(\mathbf{z})\big\| \cdot \big\|\mathbf{f}^r(\mathbf{z})\big\| \\ &\quad + (1 - \alpha)\big\|(\mathbf{I} - \mathbf{J}(\mathbf{u}))\mathbf{f}(\mathbf{u})\big\| \\ &\quad + (1 - \alpha)\big\|\mathbf{f}(\mathbf{u}) - \mathbf{J}_{\mathrm{d}}(\mathbf{z})\mathbf{f}^r(\mathbf{z})\big\|. \end{aligned} \tag{27}$$

Supposing that $\mathbf{J}_{\mathrm{d}}$ is Lipschitz continuous and bounded, and $\mathbf{f}^r$ is bounded, the following bound of $\mathbf{e}_{\mathrm{DI}}$ is obtained by integration,

$$\begin{aligned}\|\mathbf{e}_{\mathrm{DI}}(t)\| &\lesssim \int_{t_0}^{t} \big\|\mathbf{J}_{\mathrm{e}}(\mathbf{u})\mathbf{f}(\mathbf{u}) - \mathbf{f}^r(\mathbf{z})\big\|ds + \int_{t_0}^{t} \big\|\boldsymbol{\phi}_{\mathrm{e}}(\mathbf{u}) - \mathbf{z}\big\|ds \\ &\quad + \int_{t_0}^{t} \big\|(\mathbf{I} - \mathbf{J}(\mathbf{u}))\mathbf{f}(\mathbf{u})\big\|ds + \int_{t_0}^{t} \big\|\mathbf{f}(\mathbf{u}) - \mathbf{J}_{\mathrm{d}}(\mathbf{z})\mathbf{f}^r(\mathbf{z})\big\|ds, \end{aligned} \tag{28}$$

where $\lesssim$ denotes inequality up to constant factors. Therefore, combining Eqs. (18), (21) and (28) gives the full error bound of the tLaSDI approximation

$$
\begin{aligned}
\|\mathbf{e}(t)\| = \|\mathbf{e}_{\mathrm{AE}}(t) + \mathbf{e}_{\mathrm{DI}}(t)\| \\
\lesssim \int_{t_0}^{t} \|\boldsymbol{\phi}_{\mathrm{e}}(\mathbf{u}) - \mathbf{z}\| ds + \|\mathbf{e}_{\mathrm{AE}}(t_0)\| + \|\mathbf{e}_{\mathrm{AE}}(t)\| \\
+ \int_{t_0}^{t} \|(\mathbf{I} - \mathbf{J}(\mathbf{u}))\dot{\mathbf{u}}\| ds \\
+ \int_{t_0}^{t} \|\mathbf{J}_{\mathrm{e}}(\mathbf{u})\dot{\mathbf{u}} - \mathbf{f}^r(\mathbf{z})\| + \|\dot{\mathbf{u}} - \mathbf{J}_{\mathrm{d}}(\mathbf{z})\mathbf{f}^r(\mathbf{z})\| ds.
\end{aligned}
\tag{29}
$$

To relate this to the training loss (Eq. (6)) in tLaSDI, the following error terms can be defined

$$
\begin{aligned}
\epsilon_{\mathrm{int}}(t; t_0) &= \int_{t_0}^{t} \|\boldsymbol{\phi}_{\mathrm{e}}(\mathbf{u}) - \mathbf{z}\| ds \\
\epsilon_{\mathrm{rec}}(t; t_0) &= \|\mathbf{e}_{\mathrm{AE}}(t_0)\| + \|\mathbf{e}_{\mathrm{AE}}(t)\| \\
\epsilon_{\mathrm{Jac}}(t; t_0) &= \int_{t_0}^{t} \|(\mathbf{I} - \mathbf{J}(\mathbf{u}))\dot{\mathbf{u}}\| ds \\
\epsilon_{\mathrm{mod}}(t; t_0) &= \int_{t_0}^{t} \|\mathbf{J}_{\mathrm{e}}(\mathbf{u})\dot{\mathbf{u}} - \mathbf{f}^r(\mathbf{z})\| + \|\dot{\mathbf{u}} - \mathbf{J}_{\mathrm{d}}(\mathbf{z})\mathbf{f}^r(\mathbf{z})\| ds.
\end{aligned}
\tag{30}
$$

It is clear that the method error is now bounded as

$$
\|\mathbf{e}(t)\| \lesssim \epsilon_{\mathrm{int}}(t; t_0) + \epsilon_{\mathrm{rec}}(t; t_0) + \epsilon_{\mathrm{Jac}}(t; t_0) + \epsilon_{\mathrm{mod}}(t; t_0),
\tag{31}
$$

which are continuous counterparts to the loss terms defined previously. For example, applying the trapezoidal rule to approximate the integration within a small time interval $[t_n, t_{n+1}]$ gives the following approximation to the dynamics integration error,

$$
\begin{aligned}
\epsilon_{\mathrm{int}}(t_{n+1}; t_n) = \int_{t_n}^{t_{n+1}} \|\boldsymbol{\phi}_{\mathrm{e}}(\mathbf{u}) - \mathbf{z}\| ds &\approx \frac{\Delta t}{2} \|\boldsymbol{\phi}_{\mathrm{e}}(\mathbf{u}(t_{n+1})) - \mathbf{z}(t_{n+1})\| \\
&\approx \frac{\Delta t}{2} \left\|\boldsymbol{\phi}_{\mathrm{e}}(\mathbf{u}(t_{n+1})) - \boldsymbol{\phi}_{\mathrm{e}}(\mathbf{u}(t_n)) - \int_{t_n}^{t_{n+1}} \mathbf{f}^r(\mathbf{z}(s)) ds\right\|,
\end{aligned}
\tag{32}
$$

which corresponds to the integration loss $\mathcal{L}_{\mathrm{int}}$ in Eq. (8). Following a similar argument, all loss terms in Eq. (6) can be recovered from the corresponding error components in Eq. (30). This error estimate establishes a theoretical upper bound on the tLaSDI approximation error, thereby providing a theoretical foundation for the training strategy based on the loss function in Eq. (6).

## A.2 Effects of Jacobian and Modeling Loss Terms

The regularization parameters (weights) in the loss function in Eq. (6) are selected to ensure balanced contributions from each loss term. In the example of 1D1V Vlasov-Poisson Equation, it is observed that the Jacobian and modeling loss terms have relatively large magnitudes, as shown in the training loss history in Fig. 9(a). To balance their influence on the total loss, $\lambda_{\mathrm{Jac}} = 10^{-9}$, and $\lambda_{\mathrm{mod}} = 10^{-6}$ are adopted.

To further investigate the effects of these loss terms on model performance, $\lambda_{\mathrm{Jac}}$ is varied from $10^{-10}$ to $10^{-3}$, while keeping the other weights fixed. An analogous study is performed for $\lambda_{\mathrm{mod}}$. As shown in Fig. 9(b), the final values of $\mathcal{L}_{\mathrm{Jac}}$ and $\mathcal{L}_{\mathrm{mod}}$ decrease monotonically with increasing $\lambda_{\mathrm{Jac}}$ and $\lambda_{\mathrm{mod}}$, respectively. This behavior is expected, as larger regularization weights impose stronger penalties on the corresponding loss terms, resulting in smaller loss values. Fig. 9(c) presents the average maximum relative error over the entire parameter space for models trained with different regularization weights. The results indicate that excessively strong or weak regularization associated with the Jacobian and modeling loss terms can adversely affect training and model generalization performance.

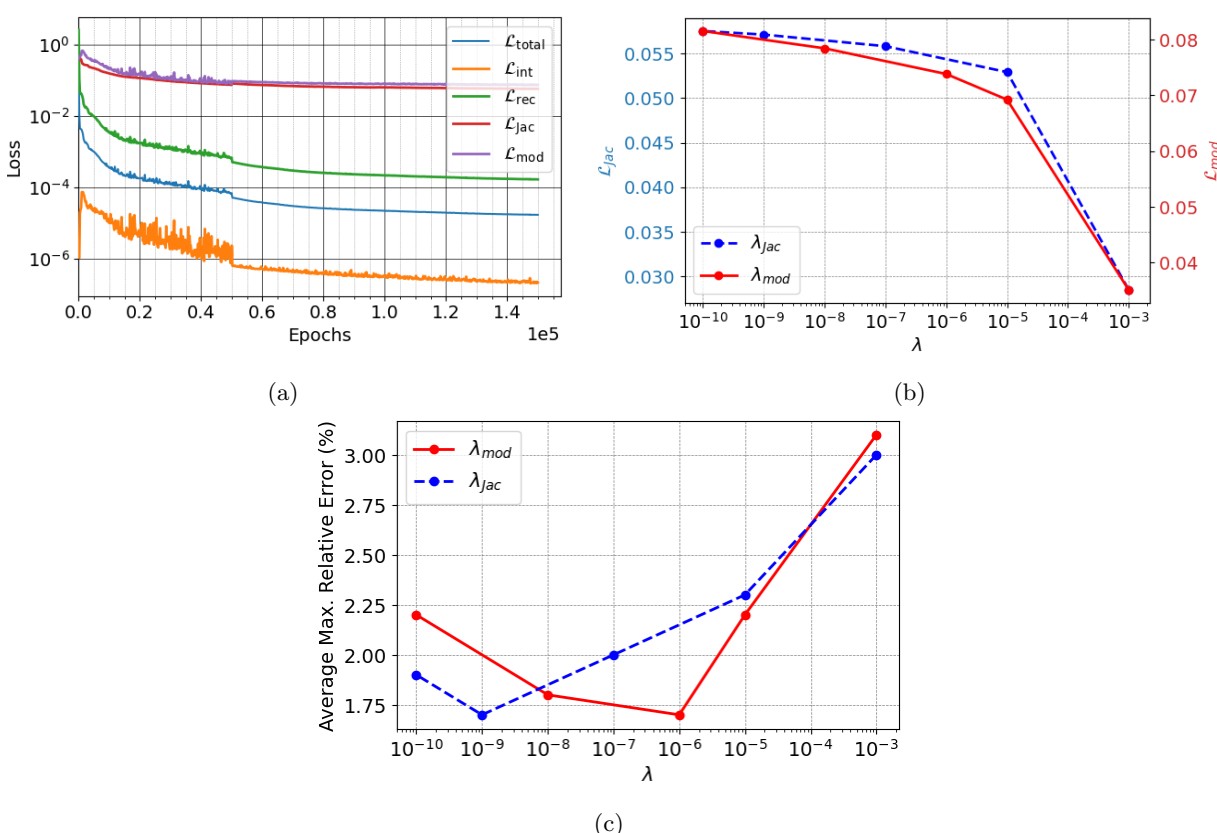

(a)

(b)

(c)

Figure 9: 1D1V Vlasov-Poisson Equation - (a) Training loss history; (b) Relationships between $\mathcal{L}_{\mathrm{Jac}}$ and $\lambda_{\mathrm{Jac}}$ and between $\mathcal{L}_{\mathrm{mod}}$ and $\lambda_{\mathrm{mod}}$; (c) Effects of the regularization parameters ($\lambda_{\mathrm{Jac}}$, $\lambda_{\mathrm{mod}}$) on prediction accuracy.

### A.3 Speed-up Analysis

For the 1D1V Vlasov-Poisson Equation, the proposed tLaSDI model achieves an average error of 1.66%, as reported in Section 3.2. The training data were generated by using high-fidelity simulations run on 4 cores of an Intel Xeon Platinum 8480 CPU. Training the tLaSDI model on an NVIDIA RTX 4000 Ada GPU takes around 45 minutes. For comparison, a single numerical simulation that has a similar error level (1.66%) with respect to the training data, produced by decreasing the spatial resolution of the high-fidelity model, requires 9.58 seconds, whereas the inference time of the trained tLaSDI model is 0.00384 seconds, yielding a $2,495\times$ speed-up.

The time step size used to generate the training data is $5 \times 10^{-3}$. Increasing the time step size of the high-fidelity simulations to $10^{-2}$ and $2 \times 10^{-2}$ while keeping the spatial resolution fixed reduces the computational time to 5.8 seconds and 5.26 seconds, respectively, and incurs small relative errors of 0.0004% and 0.014% with respect to the training data computed at the smaller time step size. Further increasing the time step size leads to unstable simulations and convergence issues. When compared to the high-fidelity simulation with the largest time step size of $2 \times 10^{-2}$, the proposed tLaSDI model achieves a $1,370\times$ speed-up. Although the examples in this paper rely on simulated data generated from PDE solvers, we emphasize that the proposed tLaSDI model is equally applicable to settings involving dynamic sensor data where the governing ODEs/PDEs are not known a priori.

### A.4 pGFINN Experiments

In this section, we demonstrate the performance of the proposed pGFINN by applying it to common benchmarks: two gas containers and a two-mass thermo-mechanical system.

### A.4.1 Two Gas Containers

The first experiment of pGFINN involves two ideal gas containers separated by a movable wall that allows heat and volume exchange. The evolution of the state variables $(q, p, S_1, S_2)$ is governed by the following ODE system,

$$
\begin{cases}
\dot{q} = \frac{p}{m}, \\
\dot{p} = \frac{2}{3}\left(\frac{E_1}{q} - \frac{E_2}{2-q}\right), \\
\dot{S}_1 = \frac{\alpha}{T_1}\left(\frac{1}{T_1} - \frac{1}{T_2}\right), \\
\dot{S}_2 = \frac{\alpha}{T_2}\left(\frac{1}{T_2} - \frac{1}{T_1}\right),
\end{cases}
\tag{33}
$$

where $m$ is the wall mass, $q$ resp. $p$ are the position resp. momentum of the movable wall, $T_i = \partial E_i/\partial S_i$, and $S_i$ resp. $E_i$, $i = 1, 2$, are the entropy resp. energy of the two subsystems, determined from the Sackur-Tetrode equation (Schroeder, 2020) for ideal gases, $S_i/Nk_B = \ln\left(\hat{c}V_i E_i^{3/2}\right)$. Here, $V_1 = q$ and $V_2 = 2 - q$ are the volumes of the containers, $N$ is the number of gas particles, and $k_B$ is the Boltzmann constant. In this experiment, $Nk_B = \hat{c} = 1$ is adopted. The total entropy of the system is $S = S_1 + S_2$, and the total energy of the system is $E = p^2/(2m) + E_1 + E_2$. This prior knowledge can be embedded into pGFINN by defining $E_{NN} := E$ and $S_{NN} := E$, as described in Section 2.1.

We first consider a system parametrized by $\alpha \in [1, 50]$. The training data is generated using the fixed initial condition $(0.87, 0.44, 1.00, 1.60)$ with $m = 1$, while varying $\alpha$ within the specified parameter range. Fig. 10(a) compares the maximum relative errors of pGFINN trained with varying numbers of uniformly distributed sampling points in the parameter space. Increasing the number of training points from 2 to 7 reduces the error from 17.0% to 3.3%, demonstrating the effectiveness of pGFINN parameterization. However, the model continues to exhibit larger errors in regions with smaller $\alpha$ values, and further reducing these errors using uniform sampling would require substantially more training points.

On the other hand, by integrating pGFINN with the physics-informed active learning strategy described in section 2.3, the pGFINN model intelligently selects training points in high-error regions. As shown in Fig. 10(b), where red circles resp. blue stars denote training points selected via uniform resp. adaptive sampling, the maximum relative error is significantly reduced to 1.2% with adaptive sampling, compared to 3.3% with

uniform sampling. Moreover, the distribution of errors over the range of $\alpha$ is substantially homogenized with active learning, ensuring that no values of $\alpha$ are unexpectedly erroneous relative to the others. Notably, the active learning strategy achieves this improvement using only 6 training points, fewer than the 7 required by uniform sampling, highlighting its efficiency and effectiveness.

As another experiment, we increase the complexity of the two gas container system's parameterization by fixing $\alpha = 10$ and considering $m \in [0.5, 5]$, which has a nonlinear relationship with the state variables. Fig. 10(c) shows that pGFINN with adaptive sampling achieves a significant reduction in the maximum relative error, from 12.8% (with 9 uniformly distributed training points) to 1.9%, using only 8 adaptively selected sampling points. Again, it can be seen that the errors are dramatically uniformized with adaptive sampling, illustrating its strong performance even in this more difficult case.

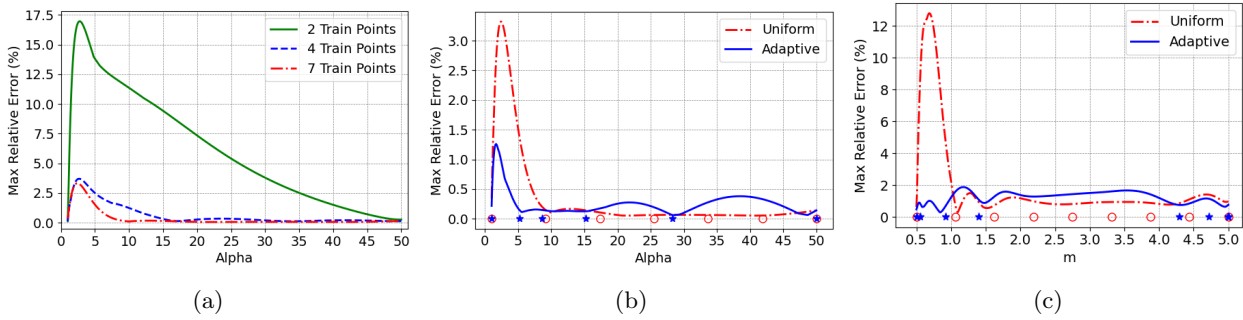

Figure 10: Two gas containers - (a) Maximum relative errors for $\alpha$ parameterization with varying numbers of uniformly distributed training points; (b) and (c) comparison of the maximum relative errors between uniform and adaptive sampling for $\alpha$ and $m$ parameterization, respectively. Red circles and blue stars denote training points selected via uniform and adaptive sampling, respectively.

We further evaluate pGFINN under 2D parameterization by considering both $\alpha \in [5, 20]$ and $m \in [0.5, 3]$. As shown in Fig. 11, pGFINN with just 14 adaptively selected training points significantly reduces the maximum relative error to 3.2%, compared to 21.1% with 25 uniformly distributed training points. Notice that the red vertical band in (a) of parameter combinations with high maximum relative error is effectively mitigated in (b) despite using many fewer training points, further highlighting the effectiveness and advantage of the active learning strategy described in Section 2.3.

### A.4.2 Two-Mass Thermo-Mechanical System

To continue testing pGFINN, we further increase the complexity of our experiments by considering a two-mass thermo-mechanical system with damping and heat exchange. The evolution of the state variables $(q_1, q_2, p_1, p_2, S_1, S_2)$ is governed by the following ODE system,

$$
\begin{cases}
\dot{q}_1 = \frac{p_1}{m_1}, \\
\dot{q}_2 = \frac{p_2}{m_2}, \\
\dot{p}_1 = -k(q_1 - q_2) - \alpha\left(\frac{p_1}{m_1} - \frac{p_2}{m_2}\right), \\
\dot{p}_2 = -k(q_2 - q_1) + \alpha\left(\frac{p_1}{m_1} - \frac{p_2}{m_2}\right), \\
\dot{S}_1 = \frac{\alpha}{2T_1}\left(\frac{p_1}{m_1} - \frac{p_2}{m_2}\right)^2 + \beta\left(\frac{1}{T_2} - \frac{1}{T_1}\right), \\
\dot{S}_2 = \frac{\alpha}{2T_2}\left(\frac{p_1}{m_1} - \frac{p_2}{m_2}\right)^2 + \beta\left(\frac{1}{T_1} - \frac{1}{T_2}\right),
\end{cases}
\tag{34}
$$

where $m_i$, $i = 1, 2$, denotes the $i$-th mass, $q_i$ resp. $p_i$ are the position resp. momentum of the $i$-th mass, and $E_i$ resp. $S_i$ are the energy resp. entropy of the $i$-th mass, with $E_i = c_i S_i$ and $T_i = \partial E_i / \partial S_i = c_i$. Further, $k$ is the spring constant between masses, $\alpha$ is the damping coefficient, and $\beta$ is the heat conductivity coefficient. The total entropy of the system is $S = S_1 + S_2$, and the total energy of the system is $E = p_1^2/(2m_1) + p_2^2/(2m_2) + k/2(q_1 + q_2)^2 + E_1 + E_2$. The following experiments adopt the choices $m_i = c_i = \beta = 1$, $i = 1, 2$.

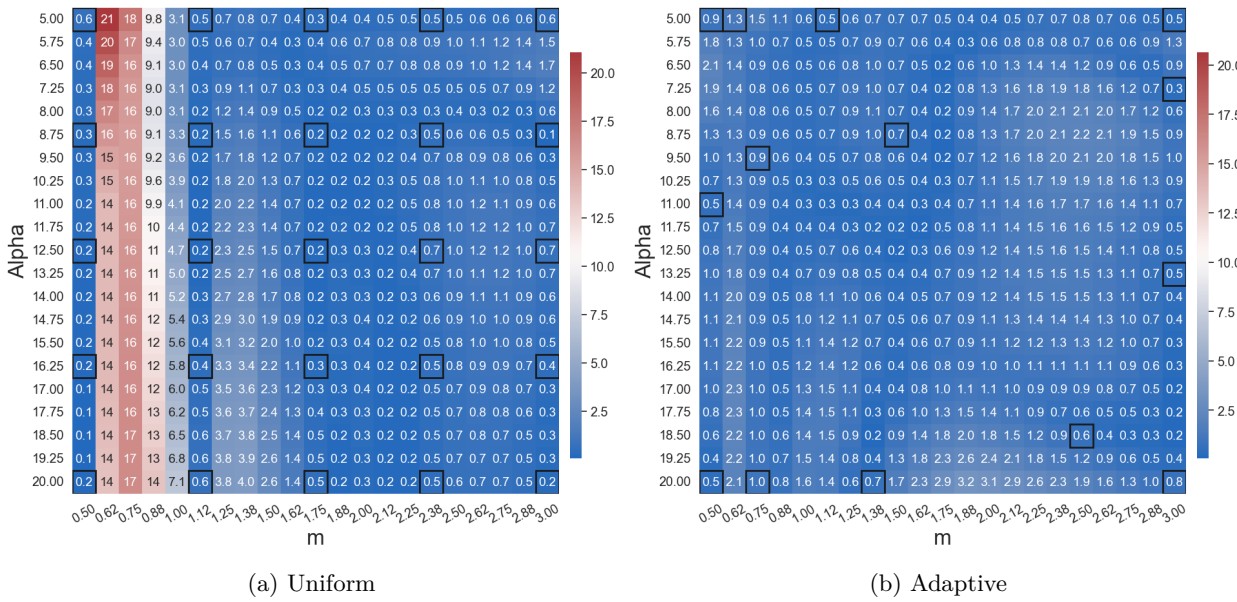

Figure 11: Two gas containers - Comparison of the maximum relative errors across the parameter space between (a) uniform sampling with 25 training points and (b) adaptive sampling with 14 training points. The training points are highlighted with black boxes.

We first consider a system parametrized by $\alpha \in [0.1, 1]$. The training data is generated using the fixed initial condition $(4.98, 0.04, 0, 9.96, 1.93, 1.92)$ with $k = 10$, while varying $\alpha$ within the specified parameter range. As shown in Fig. 12(a), increasing the number of training points from 2 to 8 reduces the error from 53.9% to 8.5%, again illustrating the impact of the pGFINN parameterization. However, uniform sampling suffers from the same drawback as before, and the model continues to exhibit larger errors in regions with smaller $\alpha$ values, which correspond to stronger oscillatory dynamics. Further reducing these errors using uniform sampling would require substantially more training points.

In contrast, Fig. 12(b) shows that pGFINN with active learning intelligently selects training points in high-error regions. With only 7 adaptively selected training points, the proposed adaptive sampling procedure reduces the maximum relative error to 4.1%, compared to 8.5% achieved by uniform sampling with 8 points.

Fixing $\alpha = 0.1$, we then consider a different parameter $k \in [0.5, 30]$ . Fig. 12(c) shows that pGFINN with adaptive sampling achieves a significant reduction in the maximum relative error in this case as well, from 2.8% (with 8 uniformly distributed training points) to 1.0% using only 7 adaptively selected points. Moreover, it can be seen that adaptive sampling has again substantially uniformized the maximum errors over the parameter range, guaranteeing consistent dynamical predictions regardless of the value of $k$.

We further evaluate pGFINN under 2D parameterization by considering both $\alpha \in [0.1, 1]$ and $k \in [0.5, 10]$. As shown in Fig. 13, pGFINN with 28 adaptively selected training points significantly reduces the maximum relative error to 4.4%, compared to 23.6% using uniform sampling with 30 training points. Similarly to the two gas containers case, the horizontal band in (a) of high maximum errors has been eliminated in (b) due to the proposed active learning strategy. This result further demonstrates the effectiveness of this procedure in improving the generalization performance of pGFINN dynamical predictions to unseen parameter configurations.

## A.5 Implementation Details

This section summarizes the implementation details of the experiments presented in Section 3.1, 3.2, and 3.3. The high-fidelity simulations were performed on an IBM Power9 CPU with 128 cores and 3.5 GHz. All training and testing in the experiments were conducted using an NVIDIA A100 GPU with 40GB of memory.

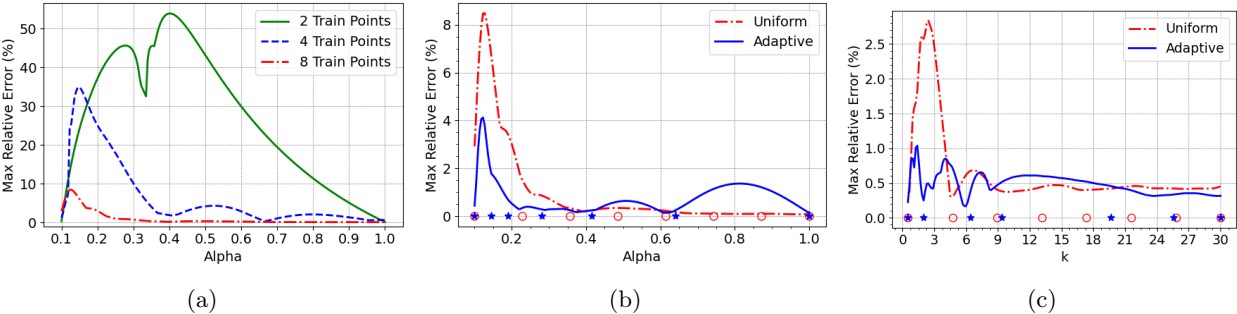

Figure 12: Two-mass thermo-mechanical system - (a) Maximum relative errors for $\alpha$ parameterization with varying numbers of uniformly distributed training points; (b) and (c) comparison of the maximum relative errors between uniform and adaptive sampling for $\alpha$ and $k$ parameterization, respectively. Red circles and blue stars denote training points selected via uniform and adaptive sampling, respectively.

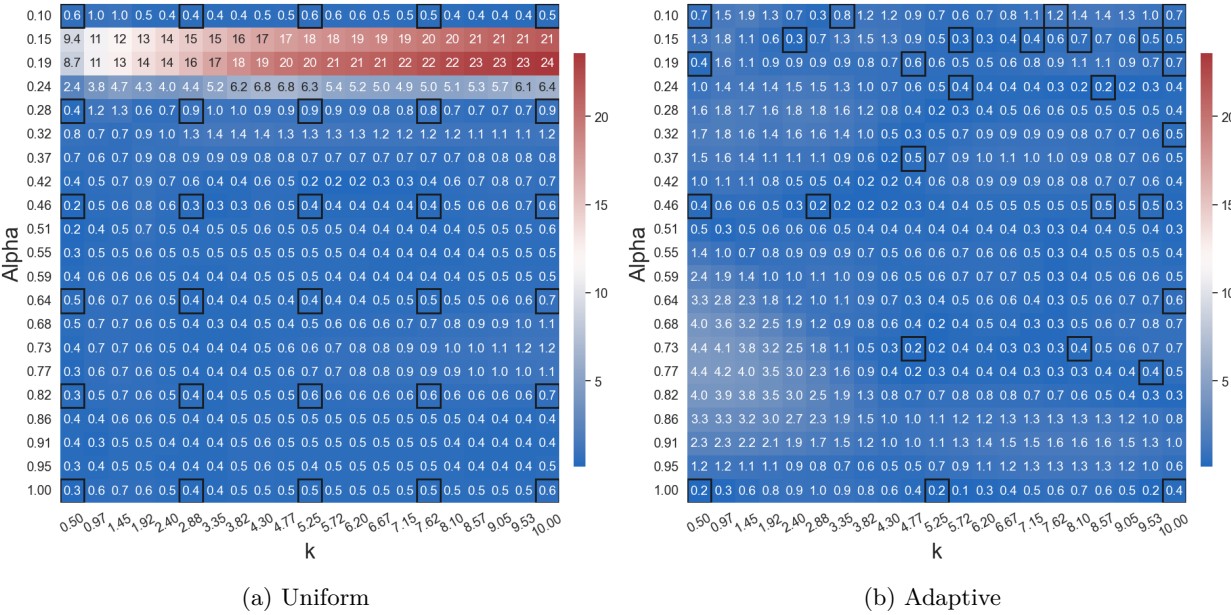

Figure 13: Two-mass thermo-mechanical system - Comparison of the maximum relative errors across the parameter space between (a) uniform sampling with 30 training points and (b) adaptive sampling with 28 training points. The training points are highlighted with black boxes.

The Adam optimizer (Kingma & Ba, 2014) from PyTorch (Paszke et al., 2019) was used for model training. All codes and data to regenerate the results in this paper can be found in the GitHub repository (`https://github.com/xiaolong7/pGFINN-tLaSDI`). This research makes use of the following open-source assets: tLaSDI (Park et al., 2024) (MIT license, `https://github.com/pjss1223/tLaSDI`), fourier neural operator (MIT license, `https://github.com/raj-brown/fourier_neural_operator`), deeponet-fno (Lu et al., 2021; Li et al., 2020; Lu et al., 2022) (CC BY-NC-SA 4.0 license, `https://github.com/lu-group/deeponet-fno`), and HyPar (MIT license, `https://github.com/debog/hypar`).

### A.5.1 One-dimensional Burgers' Equation

This section summarizes the implementation details corresponding to the results in Section 3.1.

**Data Generation**  High-fidelity simulations employ a uniform spatial discretization with nodal spacing $\Delta x = 6/1000$ and the backward Euler method with $\Delta t = 10^{-3}$ for time integration. The solution data are generated using an in-house numerical solver and the derivative of the data is computed using a backward difference scheme. The training data is subsampled from this dataset to have a spatial dimension of 200 and a temporal dimension of 201.

**Model Architectures**  The encoder adopts a 200-100-5 architecture with ReLU activation functions, while the decoder employs a symmetric structure. The hypernetwork contains 3 layers, with 20 neurons in each hidden layer and hyperbolic tangent activation. All NNs in GFINN and pGFINN contain 5 layers, with 40 neurons in each hidden layer and hyperbolic tangent activation. The forward Euler method is employed for the time integration of latent dynamics.

**Training**  The training is conducted using the loss function in Eq. (6), with regularization parameters set as $\lambda_{\text{rec}} = 10^{-1}$, $\lambda_{\text{Jac}} = 10^{-9}$, and $\lambda_{\text{mod}} = 10^{-7}$. The initial learning rate is set to $10^{-4}$ and is decayed by a factor of 0.99 every 2,000 epochs. The models are trained for a total of 15,000 epochs with a batch size of 50. The tLaSDI model, integrated with the physics-informed active learning strategy, uses an update epoch $N_{up} =$3,000, i.e., adaptive sampling is performed every $N_{up}$ epochs. Training begins with 4 parameter points located at the corners of the parameter space, and 4 additional parameter points are adaptively selected during training.

### A.5.2 1D1V Vlasov-Poisson Equation

This section summarizes the implementation details corresponding to the results in Section 3.2 and 3.3.

**Data Generation**  High-fidelity simulations employ a WENO spatial discretization ($64 \times 64$) (Jiang & Shu, 1996) and a fourth-order Runge-Kutta explicit time integration scheme with $\Delta t = 5 \times 10^{-3}$. The solution data are generated using the HyPar solver (Ghosh, 2025) and the derivative of the data is computed using a backward difference scheme. The training data is subsampled from this dataset to have a spatial dimension of 1,024 and a temporal dimension of 251.

**tLaSDI Architecture and Training**  The encoder adopts a 1,024-200-100-5 architecture with ReLU activation functions, while the decoder employs a symmetric structure. The hypernetwork contains 3 layers, with 20 neurons in each hidden layer and hyperbolic tangent activation. All NNs in GFINN and pGFINN contain 5 layers, with 40 neurons in each hidden layer and hyperbolic tangent activation. The forward Euler method is employed for the time integration of latent dynamics.

The training is conducted using the loss function in Eq. (6), with regularization parameters set as $\lambda_{\text{rec}} = 10^{-1}$, $\lambda_{\text{Jac}} = 10^{-9}$, and $\lambda_{\text{mod}} = 10^{-6}$. The initial learning rate is set to $10^{-4}$ and is decayed by a factor of 0.99 every 2,000 epochs. The tLaSDI models are trained for 50,000 epochs with a batch size of 50, followed by an additional 100,000 epochs with a batch size of 400.

**DeepONet Architecture and Training**  In the benchmark comparison (Section 3.3), DeepONet consists of a branch network and a trunk network, each with a 1,024-256-256-168 architecture and ReLU activations.

The model is trained using a mean squared error loss over 200,000 epochs, with an initial learning rate of $10^{-3}$ and weights initialized via the Glorot (Xavier) normal scheme. The learning rate is scheduled to decay according to an inverse time decay rule: $10^{-3}/(1 + 10^{-4} \times \text{iteration})$.

**FNO-3D Architecture and Training** In the benchmark comparison (Section 3.3), FNO-3D is configured with 11 channels, retains the first 5 Fourier modes, and uses ReLU activations. The model is trained using a relative $l_2$ loss over 1,000 epochs, with an initial learning rate of $2.5 \times 10^{-3}$ that is halved every 100 epochs. The input data is normalized by using unit Gaussian scaling and a weight decay regularization of $10^{-4}$ is applied to the loss function.

