# OpenReview forum: "Thermodynamically Consistent Latent Dynamics Identification for Parametric Systems"
_TMLR — Accepted by TMLR_

### Review · Reviewer_91jd · 2025-11-11

**Summary Of Contributions:**

The paper proposes a regularization scheme for the latent space of an autoencoder that encourages compliance with the domain knowledge of thermodynamics. This scheme consists in:
-an active learning method for focusing relevant training points instead of randomly sampling uniformly
-two regularization functions based on physically relevant constraints
-latent dynamics that are compliant with domain knowledge (physics) and provide interpretability?

**Audience:**

Yes

**Audience Explanation:**

This paper should be relevant for the PINNs community.

**Broader Impact Concerns:**

None.

**Claims And Evidence:**

Yes

**Claims Explanation:**

First of all, I want to mention that I am not an expert of PINNs nor in physics, making my competencies in evaluating this paper quite limited.

Nonetheless:
-the model's performance is demonstrated by the good reconstruction quality shown in table 1.
-the evaluation is accompanied by a relevant ablation study showing the positive effect of the adaptive sampling.

I think that these should be extended to show the effect of the regularization terms on the latent representation's compliance with the physical constaints. How do the weights of the jacobian and modeling error affect the outcome? Does the reconstruction become better on average, but sees more anomalies? This would be crucial to measure the impact of that physical information.

**Requested Changes:**

Deep study of the isolated impact of each element (active learning is treated, but what about the regularization scheme?).

---

> ### Author Response · Authors · 2025-12-04
> **Effect of Regularization Weights on Training and Model Performance**
>
> We thank the reviewer for the valuable comments, which have been carefully addressed in the revised manuscript. All edits are marked in red. The following discussion has been added to the Appendix A.2 of the revised manuscript.
>
> The regularization parameters (weights) in the loss function in Eq. (6) are selected to ensure balanced contributions from each loss term. In the example of the 1D1V Vlasov-Poisson Equation, it is observed that the Jacobian and modeling loss terms have relatively large magnitudes, as shown in the training loss history in Fig. 9(a). To balance their influence on the total loss, $\lambda_{\text{Jac}}=10^{-9}$, and $\lambda_{\text{mod}}=10^{-6}$ are adopted.
>
> In light of the reviewer’s comments, we have further investigated the effects of these loss terms on model performance by varying $\lambda_{\text{Jac}}$ is varied from $10^{-10}$ to $10^{-3}$, while keeping the other weights fixed. An analogous study is performed for $\lambda_{\text{mod}}$.
> As shown in Fig. 9(b), the final values of $L_{\text{Jac}}$ and $L_{\text{mod}}$ decrease monotonically with increasing $\lambda_{\text{Jac}}$ and $\lambda_{\text{mod}}$, respectively.
> This behavior is expected, as larger regularization weights impose stronger penalties on the corresponding loss terms, resulting in smaller loss values.
> Fig. 9(c) presents the average maximum relative error over the entire parameter space for models trained with different regularization weights.
> The results indicate that excessively strong or weak regularization associated with the Jacobian and modeling loss terms can adversely affect training and model generalization performance.

---

### Review · Reviewer_KFzk · 2025-11-15

**Summary Of Contributions:**

In this work the authors explore improvements to a physics aware scheme for dimensionality reduction of ODE (and therefore discretized PDE) systems. The method combines an autoencoder setup with a neural network approximation scheme that respects the first two laws of thermodynamics (pGFINN). Part of their contribution is showing that a simple autoencoder setup is sufficient.

The authors define a multi-component loss for this setup. They also define an active learning scheme to improve training performance. They then benchmark this approach on 1-D Berger's Equation and Vlasov-Poisson, for ~ 200 steps. They show that the relative error is around 1%, and show that this is better than previous methods. The model also trains faster than the approach it is directly derived from.

**Additional Comments:**

I am not an expert in the area of PDE approximation; the benchmarks look good to me but I will defer to more knowledgeable reviewers on this point.

**Audience:**

Yes

**Audience Explanation:**

The area of PDE approximation via dimensionality reduction + NN based function approximation is active, and this paper will be interesting to people in that field. The paper is also well written and can serve as an introduction to the basic approaches, particularly methods to put physical constraints/symmetries/conservation laws into the model.

**Broader Impact Concerns:**

No.

**Claims And Evidence:**

Yes

**Claims Explanation:**

To the best of my ability to check and understand them, the experiments are sound. The text in the paper is supported by the figures and tables. I am not an expert in this area so I will defer to other reviewers on whether or not the chosen benchmarks are the correct ones.

**Requested Changes:**

Does the 3000x speedup include the training time of the model? Alternatively, how does the training time of the model compare to single simulations? This information is important to understand the scenarios where one may want to deploy this method --- e.g. if training is expensive, then this method would be less useful for projects where many different models have to be tested and compared with each other.

Related to the question of speedup: what is the speedup compared to a standard ODE/PDE iteration scheme with similar error properties? Numerical integration can be sped up by choosing larger discretization in space or time, at the cost of more errors. What is the runtime of the proposed method compared to a traditional method with more coarse timesteps? To me this is an important comparison to properly position the method against classical approaches. **I think this comparison is vital for the paper.**

How does the proposed method work on more chaotic systems? For example KdV? In this setting the relative error may be larger, but it would be interesting to know if the macroscopic physical phenomena are still preserved (e.g. soliton solutions). An example like this might make the paper stronger.

---

> ### Author Response · Authors · 2025-12-04
> **Speed-up Analysis**
>
> We thank the reviewer for the valuable comments, which have been carefully addressed in the revised manuscript. All edits are marked in red. The following discussion has been added to the Appendix A.3 of the revised manuscript.
>
> For the 1D1V Vlasov-Poisson Equation, the proposed tLaSDI model achieves an average error of 1.66%, as reported in Section 3.2.
> The training data were generated by using high-fidelity simulations run on 4 cores of an Intel Xeon Platinum 8480 CPU.
> Training the tLaSDI model on an NVIDIA RTX 4000 Ada GPU takes around 45 minutes.
> For comparison, a single numerical simulation that has a similar error level (1.66%) with respect to the training data, produced by decreasing the spatial resolution of the high-fidelity model, requires 9.58 seconds, whereas the inference time of the trained tLaSDI model is 0.00384 seconds, yielding a 2,495× speed-up.
>
> The time step size used to generate the training data is $5 \times 10^{-3}$.
> Increasing the time step size of the high-fidelity simulations to $10^{-2}$ and $2 \times 10^{-2}$ while keeping the spatial resolution fixed reduces the computational time to 5.8 seconds and 5.26 seconds, respectively, and incurs small relative errors of 0.0004% and 0.014% with respect to the training data computed at the smaller time step size.
> Further increasing the time step size leads to unstable simulations and convergence issues.
> When compared to the high-fidelity simulation with the largest time step size of $2 \times 10^{-2}$, the proposed tLaSDI model achieves a 1,370x speed-up.
> Although the examples in this paper rely on simulated data generated from PDE solvers, we emphasize that the proposed tLaSDI model is equally applicable to settings involving dynamic sensor data where the governing ODEs/PDEs are not known a priori.
>
> We agree that chaotic systems are quite interesting and pose challenges for machine learning surrogates.  On the other hand, KdV is a Hamiltonian system which is non-dissipative, and therefore it is incompatible with the metriplectic physics targeted by this work. More precisely, Hamiltonian systems represent a degenerate case $M\equiv 0$ of the metriplectic formalism which necessitates a different inductive bias (e.g., HNNs).  Future work could consider metriplectic phenomena such as turbulence in fluid flow, and we expect (though are not certain) that the proposed tLaSDI model would perform well in that case also.

---

### Review · Reviewer_puUj · 2025-12-28

**Summary Of Contributions:**

This paper proposes an enhanced "thermodynamics-informed latent space dynamics identification" (tLaSDI) framework for building reduced-order models (ROMs) of parametric nonlinear dynamical systems. The core idea is to learn a low-dimensional latent representation of a high-fidelity system whose dynamics are not only accurate but also inherently obey the first and second laws of thermodynamics (energy conservation and non-decreasing entropy).

Strength:
- The motivation is clear and the paper is well-organized.
- The most significant strength is the rigorous embedding of thermodynamic priors (via the GENERIC formalism) directly into the neural network architecture. This is a substantial advantage over "black-box" operator learning methods (like DeepONet or FNO), as it guarantees physically plausible long-term behavior (e.g., stability) and provides interpretability.


Weakness:
- A potential limitation for real-world applicability is the reliance on the derivative data (u_dot) for constructing key terms in the loss function (L_Jac, L_mod). The paper briefly mentions that a weak-form formulation could be used for noisy data, but this is not explored in the experiments.
- The test cases, while standard, are relatively low-dimensional in their parametric and spatial complexity (2-3 parameters, 1D/1D1V problems). It remains unclear how well the method would scale to problems with higher-dimensional parameter spaces (e.g., >5 parameters) or much higher-dimensional physical states (e.g., 3D CFD simulations).
- Although Appendix A.2 briefly explores the sensitivity to the weights (λ_Jac, λ_mod) of the more complex loss terms, a more comprehensive ablation study would strengthen the paper. For instance, quantifying the individual contribution of the Jacobian loss (L_Jac) and the modeling loss (L_mod) to the final model performance would provide deeper insight into the necessity of each component.

**Audience:**

Yes

**Audience Explanation:**

The proposed tLaSDI framework replaces a complex "hyper-autoencoder" from prior work with a standard autoencoder coupled with the new pGFINN. This simplifies the parameterization, leading to reductions in model complexity, training cost, and inference time. Therefore, the findings might be helpful for other research.

**Claims And Evidence:**

No

**Claims Explanation:**

See Weakness. The major concern is how well the method would scale to problems with higher-dimensional parameter spaces (e.g., >5 parameters) or much higher-dimensional physical states (e.g., 3D CFD simulations).

**Requested Changes:**

Conduct additional experiments to clarify the weaknesses.

---

> ### Author Response · Authors · 2026-01-05
> **Clarifying Scope, Regularization, Weak Formulation, and Scalability (I)**
>
> We thank the reviewer for their valuable comments, which have given us the opportunity to clarify some important details.
>
> First, we would like to emphasize that this work showcases a proof-of-concept modeling approach to parametric dynamical systems, combining hard thermodynamical biases in the network architecture with active learning techniques in the latent space.  As such, the experiments in the paper demonstrate the method on a variety of physically interesting test cases designed to highlight key strengths and weaknesses in a computationally affordable setting.  While we certainly agree that extensions to higher-dimensional state and parameter spaces would provide interesting data and further challenge the method, we prefer to leave this task for future work, as the computational costs become nontrivial. In view of this, please consider the following combination of experiments and analysis which target the reviewer's concerns.
>
> **Influence of $\lambda_{\text{Jac}},\lambda_{\text{mod}}$.**
> The regularization parameters (weights) in the loss function Eq. (6) are selected to ensure balanced contributions from each loss term.
> In the example of 1D1V Vlasov-Poisson Equation, it is observed that the Jacobian and modeling loss terms have relatively large magnitudes, as shown in the training loss history in Fig. 9(a).
> To balance their influence on the total loss, $\lambda_{\text{Jac}}=10^{-9}$, and $\lambda_{\text{mod}}=10^{-6}$ are adopted.
> In light of the reviewer's comments, we have further investigated the effects of these loss terms on model performance by varying $\lambda_{\text{Jac}}$ from $10^{-10}$ to $10^{-3}$ while keeping the other weights fixed. An analogous study is performed for $\lambda_{\text{mod}}$.
> As shown in Fig. 9(b), the final values of $L_{\text{Jac}}$ and $L_{\text{mod}}$ decrease monotonically with increasing $\lambda_{\text{Jac}}$ and $\lambda_{\text{mod}}$, respectively.
> This behavior is expected, as larger regularization weights impose stronger penalties on the corresponding loss terms, resulting in smaller loss values.
> Fig. 9(c) presents the average maximum relative error over the entire parameter space for models trained with different regularization weights.
> The results indicate that excessively strong or weak regularization associated with the Jacobian and modeling loss terms can adversely affect training and model generalization performance.
>
> **Weak-form derivatives.**
> The key idea behind the weak-form formulation of our approach is to transform the strong-form ODE in Eq. (1) to a weak form by multiplying both sides with a continuous, compactly supported test function and integrating over its support.
> By applying integration by parts and exploiting the compact support of the test function, derivatives of the state variables are transferred to the test function, thereby stabilizing derivative computation in the presence of noisy data and eliminating the need for pointwise derivatives of the input (Messenger & Bortz, 2021a).
> Recent studies have successfully incoporated this weak-form formulation into latent-space dynamics identification frameworks to improve the robustness and accuracy of ROM in the presence of noise (Tran et al., 2024; He et al., 2025; Bonneville et al., 2024b).
> The proposed tLaSDI method can be extended in a similar manner by adopting a weak-form formulation for pGFINN, although this is beyond the scope of the present work.

---

> ### Author Response · Authors · 2026-01-05
> **Clarifying Scope, Regularization, Weak Formulation, and Scalability (II)**
>
> **High-dimensional parameter spaces.**
>  When extending the proposed approach to problems with high-dimensional parameter spaces, the primary challenge is the rapidly increasing number of training instances required to adequately sample the parameter space. While sparse sampling techniques, such as Latin hypercube sampling or Sobol sequences, can mitigate this burden by promoting uniform coverage, a systematic investigation of high-dimensional parameter spaces would still require substantially more data generation and training effort.
> Moreover, such an effort would primarily assess practical considerations like sampling efficiency and training costs, rather than fundamental properties of the method, such as dynamical stability and interpretability, which have already been demonstrated by the current experiments.
>
> **High-dimensional state spaces.**
> For problems involving high-dimensional physical states, the main consideration is scalability in terms of the network architecture and associated dimension reduction. As discussed in the Conclusion, the proposed tLaSDI framework is general and its dimension reduction mechanism is not restricted to standard autoencoders. Since the complexity of the model generally increases with the state dimension and depends strongly on the chosen compression technique, this provides an avenue for alternative dimensionality reduction approaches, such as convolutional autoencoders or linear compression techniques like the Proper Orthogonal Decomposition (POD), which can substantially improve efficiency based on the underlying physics.
> For example, POD is well suited to compressing diffusion-dominated problems, whereas convolutional autoencoders are often more effective at compressing advection-dominated systems. Similarly to the previous case, conducting additional experiments to this effect would require careful problem-specific design choices and substantially increased computational resources.
> Such experiments would primarily serve to demonstrate scalability rather than to provide new insights into the fundamental methodological contributions of this work.  Therefore, we choose to leave them for future investigation.
>
> Ultimately, we reiterate that the examples presented in the present version are sufficient to substantiate the key contributions of this work, namely the development of a pGFINN-based tLaSDI framework that enables stable, interpretable, and thermodynamically consistent reduced-order modeling.  We thank the reviewer for their suggestions and remain interested, particularly in considering higher-dimensional parameter and state spaces. We  plan to target more realistic examples in the future when time and computational resources permit further studies.

---

### Review · Reviewer_RFRH · 2025-12-30

**Summary Of Contributions:**

The Manuscript describes a physics informed structure the describe Hamiltonian (Poisson brackets) for energy conservations and symmetric propagator  for entropy production. The architecture uses an auto-encoder to find a latent representation  that has  a lower rank (dimension) to describe a higher complexity system.  This way a simpler, more robust  model can be used and at the same time the basic metriplectic (the one that keeps the thermodynamic 1s and second laws t (energy conservation 1st law) and 2nd law (positive entropy production) valid.

The architecture is validated with Burgers equation that has non-linearity forcing a shock wave that is regularized with friction and a more complex Vlasov-Poisson model for plasma physics, where the particles have repulsive interactions.  The results are promising, the code runs faster and its retains its accuracy.

**Additional Comments:**

I support acceptance of this Manuscript as being informative and interesting,  but also as a step to more complicated, higher dimensional models that would naturally require much more computing power. Machine learning is not always about who is having the biggest machines, but also about who has the good, creative ideas.

**Audience:**

Yes

**Audience Explanation:**

The Authors are providing an architecture that uses lower rank representation of a dynamic systems where basic physical laws are enforced. One can see how the neural network learns "by heart"  the principal features of the solution and its dynamics for the particular Hamiltonian and friction components rather than solving the equation in a generic manner - and as expected - this particular, "specialized" model can be described by a much smaller latent space than the degrees of freedom of the problem would indicate.

**Broader Impact Concerns:**

I do not have concerns on the ethical implications.

**Claims And Evidence:**

Yes

**Claims Explanation:**

The experiments with the architecture provide a convincing  and clear evidence by producing a lower dimensional latent representation of the essential physics of thermodynamical systems out of equilibrium driven to maximal entropy and the final equilibrium state.

The text is clear, the graphics supports the narrative, and the example cases are well chosen to illustrate the strength of the physics informed architecture.

**Requested Changes:**

I like the Manuscript as it is.

---

> ### Author Response · Authors · 2026-01-05
> **Acknowledgment of Reviewer’s Positive Assessment**
>
> We thank the reviewer for their positive and insightful evaluation of our manuscript and for clearly recognizing the novelty and effectiveness of our proposed interpretable, thermodynamically consistent reduced-order modeling framework. We are especially grateful for the support for acceptance and for the encouraging perspective on the role of creative model design in advancing scalable machine learning approaches for complex physical systems.

---

### Decision · Action_Editor_ZwBn · 2026-02-11

**Recommendation:** Accept with minor revision

**Additional Comments:**

Three reviewers are positive, highlighting the value of embedding thermodynamic structure and the strong empirical performance. One reviewer raises concerns primarily about (i) scalability to higher-dimensional parameter/state spaces and (ii) reliance on time-derivative data for some loss terms, requesting additional clarification/positioning. The authors’ responses and added analyses (regularization-weight study; speed-up accounting and comparisons; discussion of weak-form derivatives and scope) are constructive and largely address these points.

The remaining issues are primarily clarifications and positioning rather than requests for substantial new experiments. Please make the following minor revisions before final acceptance:
1. Speed-up reporting: In the main text (not only appendix), clearly state what is included/excluded in the reported speed-ups (training vs inference) and keep the comparison against both (a) the original high-fidelity baseline and (b) the best stable/coarser-step baseline explicit, to avoid misinterpretation.
2. Data/assumption clarity: Explicitly state the current reliance on time-derivative information for the relevant loss terms and briefly summarize the weak-form alternative for noisy settings.
3. Scope and scalability: Add a concise statement clarifying the present scope (parameter dimension and physical-state dimension) and positioning the experiments as proof-of-concept benchmarks. This should be framed as a limitation/scope statement.
4. (Optional, encouraged) If code/models are available, provide a stable link and minimal reproducibility details (hardware, training time, seeds).

**Audience:**

Yes

**Audience Explanation:**

The paper is directly relevant to TMLR readers working on scientific machine learning, reduced-order modeling, latent dynamics identification, and physics/structure-informed learning. In particular, the integration of the GENERIC (metriplectic) formalism as an inductive bias for stable and interpretable surrogate dynamics, together with a practical active-learning/sampling strategy for parametric systems, is of clear interest beyond a narrow application niche. The work provides both methodological contribution and a well-motivated demonstration on standard PDE benchmarks.

**Claims And Evidence:**

Yes

**Claims Explanation:**

The submission proposes a thermodynamics-informed latent space dynamics identification framework (tLaSDI) that combines autoencoder-based dimensionality reduction with parametric GENERIC-informed neural networks (pGFINN), enforcing key thermodynamic structure (e.g., energy conservation / free-energy consistency and nonnegative entropy production) in the learned latent dynamics across the parameter space. The methodological components are clearly described, and the empirical evaluation on canonical test problems supports the central claims: the learned reduced model achieves low relative errors while enabling substantial inference-time acceleration over the full-order solver. In addition, the authors’ revision and response addresses reviewer requests by clarifying runtime/speed-up accounting and providing additional analysis of the effects of regularization weights, which strengthens the evidentiary basis. Overall, the claims are supported by convincing evidence commensurate with the scope of the paper.

---

> ### Author Response · Authors · 2026-02-22
> **Final Revised Version Submitted Following Minor Revisions**
>
> Thank you for your careful handling of our manuscript and for the helpful summary of the reviewers’ comments. We sincerely appreciate your guidance throughout the review process.
>
> We have carefully addressed all the suggested minor revisions, including clarifications on the reporting of speed-up results, underlying data assumptions, the scope and limitations of the study, and the inclusion of a stable repository link for the code and examples.
> The revised and final version of the manuscript has now been uploaded.
> Thank you again for your time and consideration.